# FROM CHILD'S PLAY TO AI: INSIGHTS INTO AUTOMATED CAUSAL CURRICULUM LEARNING

## ABSTRACT

We study how reinforcement learning algorithms and children develop a causal curriculum to achieve a challenging goal that is not solvable at first. Adopting the Procgen environments that include various challenging tasks, we found that 5- to 7-year-old children actively used their current level competence to determine their next step in the curriculum and made improvements to their performance during this process as a result. This suggests that children treat their level competence as an intrinsic reward, and are motivated to master easier levels in order to do better at the more difficult one, even without explicit reward. To evaluate RL agents, we exposed them to the same demanding Procgen environments as children and employed several curriculum learning methodologies. Our results demonstrate that RL agents that emulate children by incorporating level competence as an additional reward signal exhibit greater stability and are more likely to converge during training, compared to RL agents that are solely reliant on extrinsic reward signals for game-solving. Curriculum learning may also offer a significant reduction in the number of frames needed to solve a target environment. Taken together, our human-inspired findings suggest a potential path forward for addressing catastrophic forgetting or domain shift during curriculum learning in RL agents.

## 1 INTRODUCTION

Humans are remarkable learners, especially when they are faced with challenging tasks. We can take advantage of curricula that are provided by teachers, and that allows us to acquire easier skills that then enable us to master more different ones. But we can also craft personalized curricula ourselves, shaping our experiences in ways that maximize our acquisition of knowledge and skills (Gopnik et al., 1999; Schulz, 2012; Schmidt et al., 2007; Loyens et al., 2008; Alfieri et al., 2011; Khan et al., 2011; Poli et al., 2020; Ten et al., 2021; Baranes et al., 2014). Similarly, reinforcement learning (RL) agents also rely on curriculum-based learning to accomplish challenging tasks (Bengio et al., 2009; Sukhbaatar et al., 2017; Florensa et al., 2018; Jabri et al., 2019; Narvekar et al., 2020).

When designing a curriculum, it is essential to strike a balance between exploitation (leveraging existing skills and information for rewards) and exploration (discovering new skills and information to enhance future decision-making). Studies have demonstrated that humans begin to master this balance between exploration and exploitation from early childhood (Gopnik et al., 1999; Kosoy et al., 2022; Sumner et al., 2019a;b; Liquin & Gopnik, 2022). The question arises: How do humans learn to reason about their own learned capabilities and use this information to bootstrap the future knowledge they need to address their current limitations?

Causal learning may be crucial for enabling us to efficiently explore various levels of task difficulty and complexity within an environment (Wang et al., 2021; Ke et al., 2021; Jiang et al., 2023). In particular, first mastering the causal relations that are involved in a simpler task can allow agents to solve more complex tasks that involve similar relations (Schölkopf et al., 2021; Florensa et al., 2018; Wang et al., 2021; Ke et al., 2021; Jiang et al., 2023). This ability contrasts with the abilities of even the most advanced RL agents. Humans can use causal inference and monitor our competence and learning progress to help guide our exploration, rather than randomly varying policies and observing the results. Causal models are well-designed precisely to afford a wide range of novel actions and interventions on the world (Pearl, 2000; Glymour et al., 1991; Spirtes et al., 2000; Peters et al., 2017). The ability to collect data from causal interventions can allow an agent to construct a new causal

model, leverage that model to make further decisions, and repeat this process for improvement. This may be key to improving the performance of RL agents in the future (Schölkopf et al., 2021; Ke et al., 2021; Wang et al., 2021; Jiang et al., 2023).

Machines may benefit from structuring their learning through a causal curriculum, improving the speed of convergence and boosting generalization by sequencing training data, developing hierarchical causal models and producing self-assessment (Bengio et al., 2009; Sukhbaatar et al., 2017; Florensa et al., 2018; Jabri et al., 2019; Narvekar et al., 2020; Soviany et al., 2022). Curriculum learning (Bengio et al., 2009; Sukhbaatar et al., 2017; Graves et al., 2017; Florensa et al., 2018; Soviany et al., 2022) is relevant and beneficial for a broad range of applications from computer vision (Shi & Ferrari, 2016; Soviany et al., 2021; 2022) and natural language processing (Li et al., 2021; Arora & Goyal, 2023) to reinforcement learning (Jabri et al., 2019; Narvekar et al., 2020; Li et al., 2023). For instance, successful learning in neural networks has resulted from a curriculum that starts small (Elman, 1993). In the case of RL, while the advantages of improving learning progress (Oudeyer et al., 2007) through a curriculum are generally recognized (Romac et al., 2021; Narvekar et al., 2020), it is still unclear how to develop or select a curriculum in the way that humans do (Graves et al., 2017). Current curriculum learning is more like following a curriculum that has been designed by a teacher than it is like designing a curriculum for yourself. Tasks are often specified a priori from human domain knowledge, and it is not necessarily clear in what sequence the tasks should be visited within a curriculum.

Our research poses a crucial question that applies to both humans and RL agents: When we approach a goal that is too complex and challenging to achieve outright, how do we evaluate our existing skills and knowledge, and then craft a curriculum of more manageable "sub-tasks"? In particular, how do we craft the tasks and actions that will allow us to build an accurate causal model that ultimately aids us in reaching our final goal? This line of work is important for both RL and cognitive science. On the cognitive science side, we seek to understand how children autonomously develop a curriculum to attain a goal that is difficult to accomplish at first. In particular, we hypothesize that children are intrinsically motivated to monitor their competence and proceed to acquire higher levels of skill accordingly, even without explicit rewards, and find evidence that this is true. By putting children and RL agents on a level playing field, we also provide a benchmark and point of comparison for human curriculum learning against curriculum learning in RL agents. On the RL front, our initial goal is to evaluate RL agents through a predefined curriculum inspired by recent research (Li et al., 2023). We subsequently use this as a baseline against which we assess RL agents that include level competence as an auxiliary reward, inspired by our results with children. Our findings indicate that incorporating this reward significantly enhances the pace of learning and improves convergence. We discover something new about children; that is, we show that children use intrinsic rewards based on level competence. We then show that designing RL agents in a similar way leads to great progress and improvement.

We outline the definitions of the terms we used in the paper here in Table 1:

| Term | Definition |
| --- | --- |
| Level Competence | An indicator of success in an episode of game play on a specific level, reaching 100% when the specific level is successfully solved in that episode (also see Table 4 in Appendix A.2 for more details of calculating competence in each game) |
| Global Competence | A measure of success in an episode of game play with respect to the target level, reaching 100% if the target level is successfully solved in that episode |
| Level Advancement | Difference between current level competence and initial level competence within a particular level in the curriculum; it is a measure of how much competence increases or decreases in the same level |
| Global Advancement | Difference between current global competence and initial global competence across levels in the curriculum; it is a measure of how much competence increases or decreases with respect to achieving the target level |
| Auxiliary Reward | An internal reward used by the RL agent that augments the extrinsic reward |

Table 1: Definitions of the terms we used in this paper

## 2  RELATED WORKS

Our work relates to prior work in cognitive science, curriculum learning, and active causal learning.

**Automated Learning in Children** Related work in cognitive science suggests that child learning resonates with the Goldilocks principle: children opt for information that is neither too easy nor too complex, but "just right" and moderately predictable (Cubit et al., 2021; Kidd et al., 2012; 2014). Furthermore, children as learners seem capable of monitoring the "zone of proximity" between their current capabilities and the goal at hand; enabling them to progress from what they cannot do to what they can learn to do with the interventions of an adult or a teacher (Vgotsky, 1987). Infants allocate their visual attention based on surprise, predictability and learning progress of the environmental structure (Poli et al., 2020). 4- to 6-year-old children use their improvement over time to decide whether to persist on a challenging goal on their own (Leonard et al., 2023). And by age 7, children ask questions that yield higher information gain when problems are more difficult (Ruggeri & Lombrozo, 2015). However, there are no systematic studies showing that young children can indeed construct an appropriate curriculum in order to master complex goal-directed tasks, nor studies that would allow a comparison between children and artificial agents. In this work, we discover something new about children; that is, we show that children use intrinsic rewards based on level competence. We then show that designing RL agents in a similar way leads to substantial improvement.

**Curriculum Learning** Bengio et al. (2009) introduced the concept of curriculum learning and proposed that an effective approach provides examples that strike a balance – neither overly simple nor excessively challenging. This concept is further supported by theoretical proofs in reinforcement learning (Li et al., 2023). Various metrics have been proposed to measure task difficulty, including the transferability of models trained on other tasks to the current task (Weinshall et al., 2018), complexity-driven progress, and loss-driven progress (Graves et al., 2017). Sukhbaatar et al. (2017) proposed a framework for automatic curriculum learning through self-play, while Florensa et al. (2018); Baranes et al. (2014) proposed a framework utilizing automatic goal generation. Dennis et al. (2020) introduced the Unsupervised Environment Design paradigm, formalizing variation in environments by parameters, along with the PAIRED algorithm that produced an implicit curriculum. Prioritized Level Replay (Jiang et al., 2021a) also implicitly learns a curriculum for training an RL agent not through environment adaptation, but through judicious selection of past levels to replay based on learning potential. Later, Jiang et al. (2021a) unified these two concepts under the Dual Curriculum Design framework with Robust PLR as a representative algorithm, and Parker-Holder et al. (2022) introduced ACCEL, an evolutionary-based approach for editing environments to form a curriculum.

Active causal learning may also contribute to RL curriculum development. In RL curriculum development, we perform environmental interventions to enable the swift achievement of learning objectives, resembling active causal learning (Murphy, 2001; Tong & Koller, 2001; He & Geng, 2008; Eberhardt, 2012; Squires et al., 2020; Ghassami et al., 2018; 2019; Hyttinen et al., 2013; Kocaoglu et al., 2017; Lindgren et al., 2018; Scherrer et al., 2021). However, curriculum learning in RL differs from active causal learning in that it dispenses with explicit causal structure learning and prioritizes targeted causal interventions, disregarding non-essential variables or edges.

In the current approach, we draw inspiration from child development studies. We use level competence as an auxiliary reward within a curriculum inspired by Li et al. (2021). Our results show that employing auxiliary rewards mitigates RL agent catastrophic forgetting, leading to faster and improved convergence.

## 3  EXPERIMENT DESIGN: PROCGEN ENVIRONMENTS

Procgen (Cobbe et al., 2020) represents a procedurally generated environment that develops a wide range of RL games with varying levels of difficulty. To systematically analyze curriculum learning for both human players and RL agents, we selected a subset of Procgen environments and tailored them by adjusting game difficulty along a single parameter or variable. Our experimentation focused on three distinct games: Leaper, Climber, and Heist (Figure 7 in Appendix A.3). These games were selected on two criteria: (i) they contain difficulty levels that vary along clearly defined, quantifiable axes and (ii) they are appropriate for children, striking a balance between challenge and accessibility.

The goal of each of game is provided in Table 2 in Appendix A.2. Each game includes three levels (with some participants encountering four levels), where Level 1 represents the easiest, and Level 3

represents a demanding challenge. Participants, or agents, are tasked with achieving success at the target level, which is Level 3. Level 4 was presented to a random subset of the participants. Level 4 is the most challenging level, and most importantly, the level participants do not need to necessarily learn to achieve the target Level 3. Table 3 in Appendix A.2 describes the levels for the different games.

# 4 AUTOMATED CURRICULUM LEARNING IN CHILDREN

We ask how children scaffold their own learning to solve a difficult level of a game. Our primary focus is to explore whether children employ a systematic and rational approach in constructing a learning curriculum for themselves - a question that has not previously been investigated. To assess this, we observe children engaging with Procgen games (Cobbe et al., 2020) of varying difficulty levels and analyze their automated decisions in curriculum development.

## 4.1 EXPERIMENTAL PROCEDURE

**Overview** We have gathered data from a cohort of 22 children (10 females, 12 males), with a mean age of 5.55 years ($\sigma = 0.74$ years) at public museums in the Bay Area, California, USA. Participants were told that they would be rewarded with a sticker if they won a target game level. The target level was set to be too challenging to win in a single attempt - it required some form of multi-step, curriculum-like learning. Children were then given the opportunity to autonomously select different difficulty levels of the game (Levels 1-4, with Level 3 being the target game level). Children could play either until they won the target level, or for up to a total of 10 rounds, whichever came first.

**Familiarization with the Game Environment** The Procgen games were presented to participants in a laptop computer. Participants were randomly assigned to play one of the games. Participants first underwent a familiarization phase of the game. Using a physical video-game controller that consisted of four buttons (up, down, left, right), they practiced how to navigate an empty game environment. For example, in the game Leaper, participants explored an environment without any lanes or obstacles between the starting point and the finishing line.

**Automated Curriculum Learning with an Extrinsic Goal** Next, participants were introduced to the target level of the game. They were told that they would be rewarded with a sticker if they won the target level, but they were not told any further details of the game. Since the experiment aimed to measure curriculum learning in a case where the goal was too challenging to attain outright, participants had to fail the target level before they could continue with the experiment. If a participant passed the target level, they were reassigned to play a different game. After the participant failed the target level in their first attempt, the experimenter restated to them that the goal was to win that level in order to get a sticker. To probe causal understanding of the rules and physics of the game, the experimenter then asked the participant to tell them how the game worked and what the participant had to do to get the sticker.

Then, participants were asked which level of the game they wanted to play next and why. They were presented with images of four levels that quantitatively varied along a single game axis, as shown in Figure 7 in Appendix A.3. Critically, they could see increasing number in an axis directly related to game difficulty along the levels, e.g. there were more lanes in some levels than in others in Leaper (see Table 3 in Appendix A.2), but were not explicitly told the relative difficulty of each level. Participants were presented with a total of three levels of difficulty: the target level and two levels that were incrementally easier than the target level. To probe their understanding that practicing on easier levels would lead to improvement on the difficult target level, a subset of 7 participants were further presented with an additional level (Level 4) that was even more difficult than the target level. Additional effort spent on solving Level 4 was causally irrelevant to solving the target level and winning the sticker.

After playing an episode of the game, participants received explicit verbal and visual feedback from the experimenter about whether they passed or failed the game. Participants either saw a happy face on the computer screen and learned that they passed the level, or they saw a sad face and learned that they did not pass the level. They did not receive feedback from the experimenter on *how* competent they were at the game (in the way that we computed level competence for analyses). In other words, the only way for them to determine their amount of competence was to intrinsically monitor how

far they went in an episode of playing the game. After every other episode, participants were also reminded that their goal was to solve the target level and win a sticker.

This automated curriculum learning process repeated either until participants won the target level, or for up to a total of 10 rounds, whichever came first. Those who did not pass the target level by the tenth trial were invited to play the target level again and then the experiment concluded.

## 4.2 EVALUATION METRICS

We recorded each participant's selections of the game level throughout the experiment as their curriculum as well as their reasoning. Additionally, we also measured the granular competence that participants made on each attempt or episode of the game. This was measured as a percentage, $100\%$ indicating success and $< 100\%$ indicating failure at a particular point of the game (see Table 4 in Appendix A.2 for how we compute competence for each game). We also counted the number of rounds it took for participants to win the target level of the game.

## 4.3 RESULTS

Overall, children made an average global advancement toward the goal of $\mu = 58.4\%$ with a standard error of $SE = 11.8\%$ (this was computed as the difference in competence between the initial attempt and the final attempt at the target level), which was significantly different from 0% ($t(15) = 4.96$, $p < .001$). This was even though they did not necessarily make positive level advancement between the first and last attempt at the same level ($t(17) = .29$, $p = .78$ in a one-sample t-test compared to $\mu = 0$). After failing the initial target level, 72.7% participants started their curriculum by selecting an easier level: 9 chose Level 1, 7 chose Level 2, 5 chose Level 3, 1 (of the 7 presented with Level 4) chose Level 4 - the even more difficult level.

Next, we examined the change in levels selected by children across their automated curricula, given their competence on their current level (see Figure 1). A change in level of 0 indicates that the participant chose to remain on the same level, a change in level of -1 indicates that they chose the next easiest level, and so on. We found that children were more likely to remain on the same level or go to easier levels if their progress was low, and were more likely to move to more challenging levels if their competence was high. More specifically, level change (z-scored) was positively predicted by level competence (z-scored) in a linear mixed effects model with participant as a random effect, $\beta = .37$, $t(75) = 3.38$, $p < .01$. Thus, children, like adults in free exploration (Ten et al., 2021), used competence information to avoid overly easy tasks and advance to more difficult levels that were closer to the target level in their curriculum. However, children neither used their global advancement (z-scored) to guide their level change in the curriculum (z-scored) ($\beta = .029$, $t(77) = -.26$, $p = .80$), nor did they use their level advancement (z-scored) to guide their level change (z-scored) ($\beta = .070$, $t(50) = .49$, $p = .63$). One possible

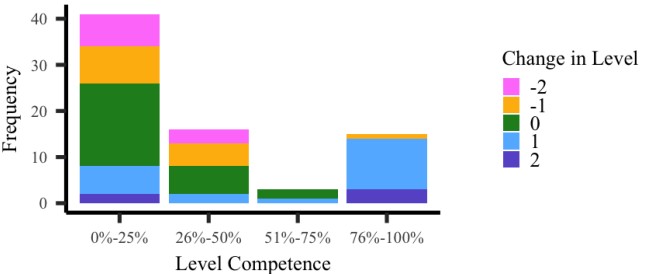

Figure 1: Level adjustments based on children's competence on the current level. The x-axis measures current level competence as a percentage; the y-axis shows subsequent level adjustment frequency. A level change of 1 implies choosing a game one level harder, while -1 means opting for one level easier. This figure includes participants' selections of levels easier than or equivalent to the target level (see Figure 9 in Appendix A.4 for the inclusion of participants selecting an extra challenging level beyond the prescribed target level). Overall, children tend to remain on the current level when their competence is less than $75\%$. However, upon reaching a $76\%$ completion rate, children often transition to more challenging levels. Conversely, when children demonstrate insufficient competence ($\leq 50\%$), they are more inclined to return to easier levels (see Figure 10 in Appendix A.4 for children's level adjustments organized by competence in each individual level). Thus, children adapt their learning trajectory based on their performance, indicating a somewhat systematic approach to curriculum development.

reason is because children were not told to freely explore all levels, but were given the extrinsic goal of solving a difficult level (Level 3). Thus, selecting more difficult levels was appealing even when children did not make much level advancement. This is evident in Figure 10 in Appendix A.4.

Furthermore, we also found that children demonstrated a causal understanding of their curriculum learning. We introduced 7 children to Level 4 which was even more difficult than the target Level 3. If children were selecting their curricula at random, all four levels should be equally likely to be selected, resulting in a 25% chance of selecting Level 4. However, children only selected Level 4 9.68% of the time. This suggests that children recognized that spending extra effort on Level 4 would not cause them to win a reward.

## 5 Hand-designed Curriculum Learning in RL Agents

### 5.1 Formulation

Reinforcement learning (RL) agents are trained to solve tasks modeled as a Markov Decision Process (MDP) (Bellman, 1957; Sutton & Barto, 2018). In this formulation, $M = \langle \mathcal{S}, \mathcal{A}, T, \mathcal{R}, \gamma \rangle$, where $\mathcal{S}$ is the state space, $\mathcal{A}$ is the action space, $T$ is the transition function, $R \in \mathcal{R} : \mathcal{S} \to \mathbb{R}$ is the reward function, and $\gamma$ is the discount factor. Within each MDP, the agent acts according to policy $\pi : \mathcal{S} \to \mathcal{A}$, which concerns low-level control. For this work, Proximal Policy Optimization (PPO) (Schulman et al., 2017) is used to train the low-level control policy $\pi$.

During the training of the low-level control policy, the agent experiences different distributions of MDPs depending on the agent's curriculum. Specifically, the distribution of MDPs is provided by a high-level curriculum selection function $\phi : \Theta \to M$, where $\Theta$ are environment parameters that specifies the "difficulty" of the tasks. Generally, tasks that are more difficult require greater sophisticated behaviors of the agent. For this work, we assume that difficulty is modulated through only state distribution changes $\mathcal{S}$; all other aspects of the MDP remain the same across task difficulty.

The goal of the agent is to learn a policy $\pi$ that can solve a difficult target task $M_t$, analogous to needing to solve the target level as specified in Sec. 3. Tasks are considered solved when the episode reward $R$ exceeds a predetermined threshold, $R_S$. For the purposes of monitoring agent learning, we assume there exists a function $\Lambda : \mathcal{S} \to \mathbb{R}$, called level competence, that describes how far the agent is into the task. Level competence is bounded from 0 to 1. Note that the agent does not necessarily have access to level competence.

### 5.2 Methods

We assess curriculum learning in RL agents through a hand-designed curriculum on the Procgen game of Leaper. Task difficulty $\Theta$ is varied within the curriculum by parameterizing the number of water lanes, from 1 (easiest) to 5 (hardest). The target task $M_t$ has 5 water lanes, consistent with Level 3 (Sec. 3). Motivated by recent work that suggests tasks should be solved in an easiest-to-hardest fashion for better sample complexity (Li et al., 2023), our hand-designed curriculum function $\phi$ starts at 1 water lane (easiest) and trains the agent until a mean episode reward of 9 (out of possible 10) is achieved. Leaper is a sparse rewards problem: a reward of 10 is only provided when an episode is solved (and 0 otherwise). Specifically, we run $w = 16$ parallel tasks, initially starting all 16 at 1 water lane. Then, the curriculum function $\phi$ increments $\Theta$ by $1/w$ (e.g., 1/16) to advance to a harder distribution of tasks. After each PPO update, the agent is evaluated against the target task $M_t$. If the agent successfully solves $M_t$, training concludes. Otherwise, training continues until a maximum number of frames $f_m = 10 \times 10^6$. The level competence $\Lambda$ for Leaper is the vertical lane that the agent has reached, normalized by the total number of lanes between the start and the finish line. We use hyperparameters from Jiang et al. (2021b) for training the policy $\pi$ using PPO.

### 5.3 Baseline Curriculum Learning Results

We conduct six trials of training the agent using the hand-designed curriculum as specified by $\phi$. Representative results are summarized in Figs. 2-3. Overall, results are poor. None of the six trials successfully finished training. All trials experienced training divergence, where the agent experiences catastrophic forgetting, leading to a permanent regression of reward to zero and a

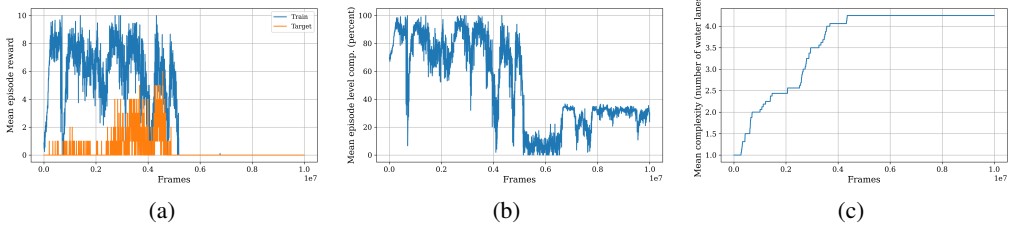

(a)                        (b)                        (c)

Figure 2: Representative results for baseline curriculum learning with an RL agent. (a) Time history of mean episode reward obtained by the agent in both the training tasks and the target task. Training divergence from catastrophic forgetting results in a regression of reward to zero, which occurs around 5.18 million frames. (b) Time history of mean level competence in the training tasks. (c) Time history of the training task difficulty, as measured by number of water lanes.

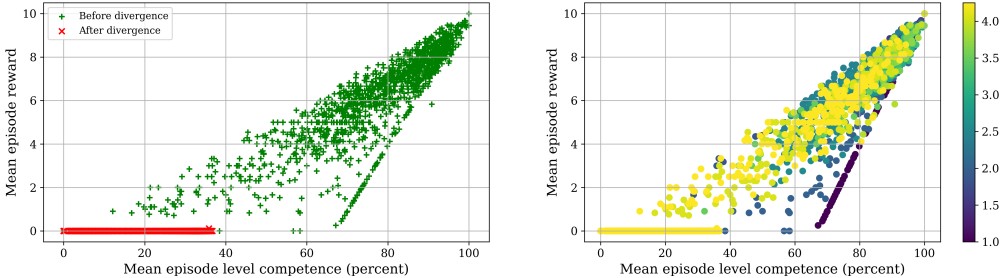

(a) Reward by level competence, colored by divergence. (b) Reward by level competence, colored by difficulty.

Figure 3: Level competence is a proxy for reward. (a) Prior to training divergence, mean episode training reward is proportional to mean episode level competence. After training divergence, this relationship no longer holds: the reward remains at zero regardless of level competence. (b) Before training divergence, the exact relationship of reward and level competence depends on the task difficulty. The easiest task (1 water lane, dark blue) has the greatest slope, since changes in level competence yield relatively greater mean training reward. The slope decreases as difficulty increases because tasks have increasingly more vertical lanes before the goal.

corresponding decrease in level competence. Figure 2 shows a representative time history of an agent exhibiting training divergence by catastrophic forgetting. It has been shown that agents can experience catastrophic forgetting in continual learning problems with distribution shifts (McCloskey & Cohen, 1989; Kirkpatrick et al., 2017; Toneva et al., 2018; Nguyen et al., 2019). In our problem, our hand-designed curriculum is inducing intentional distribution shifts in order to train the agent in tasks of increasing difficulty. Behaviorally, when this divergence arises, the agent completely forgets the ability to cross lanes, thereby losing the reward signal in such a sparse rewards problem. Note that, for the representative result in Fig. 2c, after divergence, the agent never exceeds 40% mean level competence, which is approximately where the first lane occurs in the distribution of tasks of this difficulty (4.25 water lanes). Without the reward signal, the agent does not receive feedback on which actions are good to take. Although for simpler environments it may be possible to recover, as the difficulty is smaller and random actions may find the goal, for harder environments with multiple lanes, this divergence is unrecoverable. The average number of water lanes at time of divergence is 3.6667 lanes, with the minimum being 2.6875 lanes.

Figure 3 shows the relationship between mean episode training reward and mean episode level competence. Prior to training divergence, level competence is a proxy for reward, although the specific relationship depends on the difficulty of the task.

## 5.4 CURRICULUM LEARNING WITH AUXILIARY REWARDS

Our experiments with children suggested that children used signals about their level competence to craft a curriculum and determine which level to attempt next. Inspired by this finding, we conducted an additional experiment where the agent uses level competence as an auxiliary reward. As before,

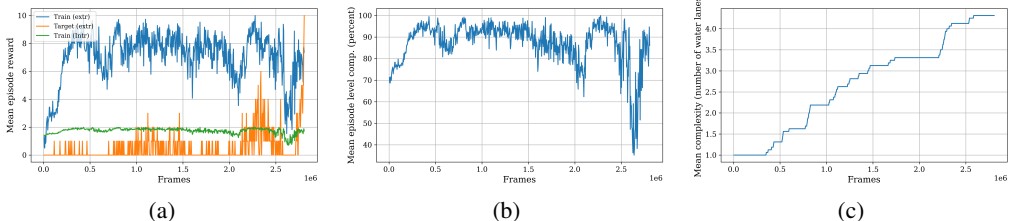

Figure 4: Representative results for curriculum learning with an RL agent while training on level competence as an auxiliary reward. (a) Time history of mean episode reward obtained by the agent in both the training tasks and the target task. The auxiliary reward used for training is also shown, which is derived from the agent's level competence. The agent begins to generalize to the target task around 2.8 million frames, eventually leading to solving the target task in 2.806 million frames. (b) Time history of mean level competence in the training tasks. (c) Time history of the training task difficulty, as measured by number of water lanes.

six trials were conducted. Specifically, the auxiliary reward function is $\mathcal{R}_i = 2\Lambda/100$, which is calculated at the termination of an episode. For this experiment, we assume the agent has access to level competence, leaving how it would be computed for future work.

When training using auxiliary rewards, results were markedly improved. Five of the six trials were able to successfully finish training by solving the target level via generalization. On average, generalization occurred at 3.353 million frames. Figure 4 shows a representative time history for training using level competence as an auxiliary reward where generalization occurred. Although the remaining trial did not solve the target environment, training reached the maximum number of frames allowed without experiencing divergence.

The experimental results with using an auxiliary reward for level competence suggest this prevents training divergence. We confirmed this by examining one particular trial, shown in Fig. 5, in which the agent *recovers* from catastrophic forgetting using the auxiliary reward of level competence. A difficulty increase from 4 to 4.0625 water lanes around 4.141 million frames induces catastrophic forgetting, eventually leading to zero (extrinsic) reward and the complete loss of ability to cross lanes around 4.5 million frames. However, unlike in the baseline experiments, the agent still obtains a reward and thus a learning signal. Although the auxiliary reward decreases with level competence, it does not reach zero. Despite being small, the learning signal is sufficient: the agent can still receive feedback about how to increase competence. This leads to a gradual restoration of agent capability, starting with an increase in level competence around 4.75 million frames. Eventually, this restoration yields a gradual increase of extrinsic reward, which is only obtained if the agent can solve the task. Although the restoration is not quick, taking about 1.5 million frames, the agent can nonetheless recover from catastrophic forgetting that would have otherwise led to training divergence.

## 5.5 ADDITIONAL EXPERIMENTS

We conduct three additional experiments: a random level selection baseline and a stochastic curriculum that samples from either random levels or previously solved levels. Results are available in App. A.5. We see that having a stochastic curriculum using previously solved levels also prevented training divergence, but it did not advance the curriculum as well as with auxiliary rewards (Sec. 5.4).

## 6 GENERAL DISCUSSION

From early childhood, humans can create a causal curriculum to reach a goal that is too challenging to attain in a single attempt. We found that 5- to-7-year-old children used level competence as an intrinsic reward signal to decide what level to play in order to eventually solve the most challenging level. On one hand, participants who were less competent in their current level were more likely to choose to play the same level again or an easier level. On the other hand, participants who were more competent at their current level were more likely to choose a more difficult level. With the granular

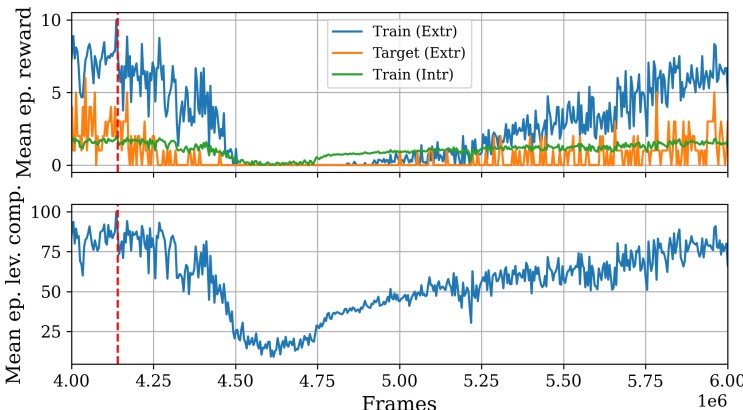

Figure 5: Training using level competence as an auxiliary reward can recover from catastrophic forgetting that would have otherwise led to training divergence. The vertical red line marks the increase in task difficulty from 4 to 4.0625 water lanes, precipitating (recoverable) catastrophic forgetting.

data of level competence, we can see exactly when participants were more likely to choose the same, an easier, or a more challenging level of the game.

## 6.1 LEVEL COMPETENCE AS AN AUXILIARY REWARD SIGNAL TO OVERCOME CATASTROPHIC FORGETTING

While children are given the extrinsic reward of a sticker if and only if they solve the target level of the game, they can visually observe their success in each game play and so assess their competence at each level. Seeing their competence allows for dense reward signals that are set by their performance. They further use this signal to determine the next step to take in the curriculum, leading to an average of 58.4% competence on the target goal at the end of their automated curriculum learning. Similarly, level competence is a crucial and beneficial signal for RL agents. Our RL training demonstrates that exploiting level competence as an auxiliary reward signal reduces the odds of divergence and catastrophic forgetting.

## 6.2 FUTURE DIRECTIONS

Building on our finding that children employed level competence as a metric to assess advancements in curriculum-based learning, our RL experiments incorporating level competence as an auxiliary signal yielded notably improved results. This compelling evidence underscores the pivotal role of level competence as an additional incentive to overcome distribution shifts induced by a curriculum. These findings suggest that extracting level competence information from high-dimensional image inputs and using it as a reward mechanism has the potential to significantly enhance the efficiency of RL agents in curriculum learning. In fact, it might potentially enable them to autonomously acquire proficiency in curriculum learning. In addition, we hope to further explore whether *automated* curriculum learning necessarily outperforms curriculum learning in a sequence based on strictly incremental difficulty or in a random sequence (Atkinson, 1972; Kornell & Metcalfe, 2006) for both children and RL agents. Finally, we are conducting an ongoing analysis of children's explanations of the nature of the game rules and their justifications for choosing particular levels. Preliminary results suggest that children's experience at simpler levels does help them build a causal model of the game, and the causal model assists them in accomplishing progress at higher levels.

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

## A APPENDIX

### A.1 PROCGEN ENVIRONMENTS

There are 12 different Procgen environments. Figure 6 taken from (Cobbe et al., 2020) showing all available Procgen environments.

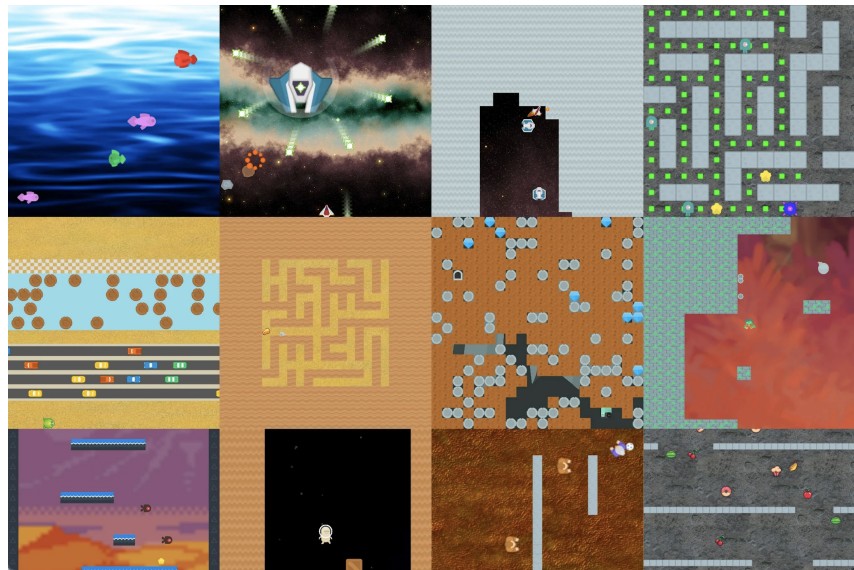

Figure 6: An illustration of all Procgen environments

### A.2 OUR ADAPTED PROCGEN ENVORIMENTS

| Game | Rule |
|------|------|
| Leaper (Log) | Cross the finish line and avoid going in the water. |
| Leaper (Car) | Cross the finish line and avoid getting hit by a car as car lanes increase. |
| Climber (Enemy) | Reach the star on the top platform and avoid enemies as number of enemies increases. |
| Climber (Platform) | Reach the star on the top platform and avoid a single enemy as number of platforms increases. |
| Heist (Size) | Reach the final jewel by getting a key and unlocking a lock as size of maze increases. |
| Heist (Key) | Reach the final jewel by getting keys and unlocking locks as number of locks and keys increase. |

Table 2: The rules to win each Procgen game are presented. Participants had to figure out these rules through their own causal curriculum learning.

| Game | Axis | Level 1 | Level 2 | Level 3 (target level) | Level 4 |
|------|------|---------|---------|------------------------|---------|
| Leaper | Log | 1 log lane | 3 log lanes | 5 log lanes | 7 log lanes |
| Leaper | Car | 1 car lane | 3 car lanes | 5 car lanes | 7 car lanes |
| Climber | Enemy | 0 enemies | 1 enemy | 2 enemies | 3 enemies |
| Climber | Platform | 1 platform | 2 platforms | 3 platforms | 4 platforms |
| Heist | Size | small | medium | large | extra large |
| Heist | Key | 1 key/lock | 2 keys/locks | 3 keys/locks | 4 keys/locks |

Table 3: The difficulty levels of the Procgen games of Leaper, Climber, and Heist are presented with Level 1 as the easiest level, Level 3 as the target level, and Level 4 as the most difficult level.

| Game | Competence Calculation |
|------|------------------------|
| Leaper | the lane that a player reached, normalized by the total number of lanes between the start and the finish line |
| Climber | the platform that a player reached, normalized by the total number of platforms between the start and the finish line |
| Heist | the number of objectives that a player has achieved (gathering a key, unlocking a lock, or gathering the jewel), normalized by the total number of objectives a player has to achieve |

Table 4: We provide the calculation of competence in an active episode of each game. Note for Heist: there are 2*L + 1 objectives, where L is the number of locks. A game with 3 locks would have 7 objectives: collect 1st key, unlock 1st lock, collect 2nd key, ..., collect jewel. Collecting the 1st key would lead to 14.2857% competence, unlocking the 1st lock would lead to 28.5714% competence, and so forth.

A.3   PROCGEN LEVELS

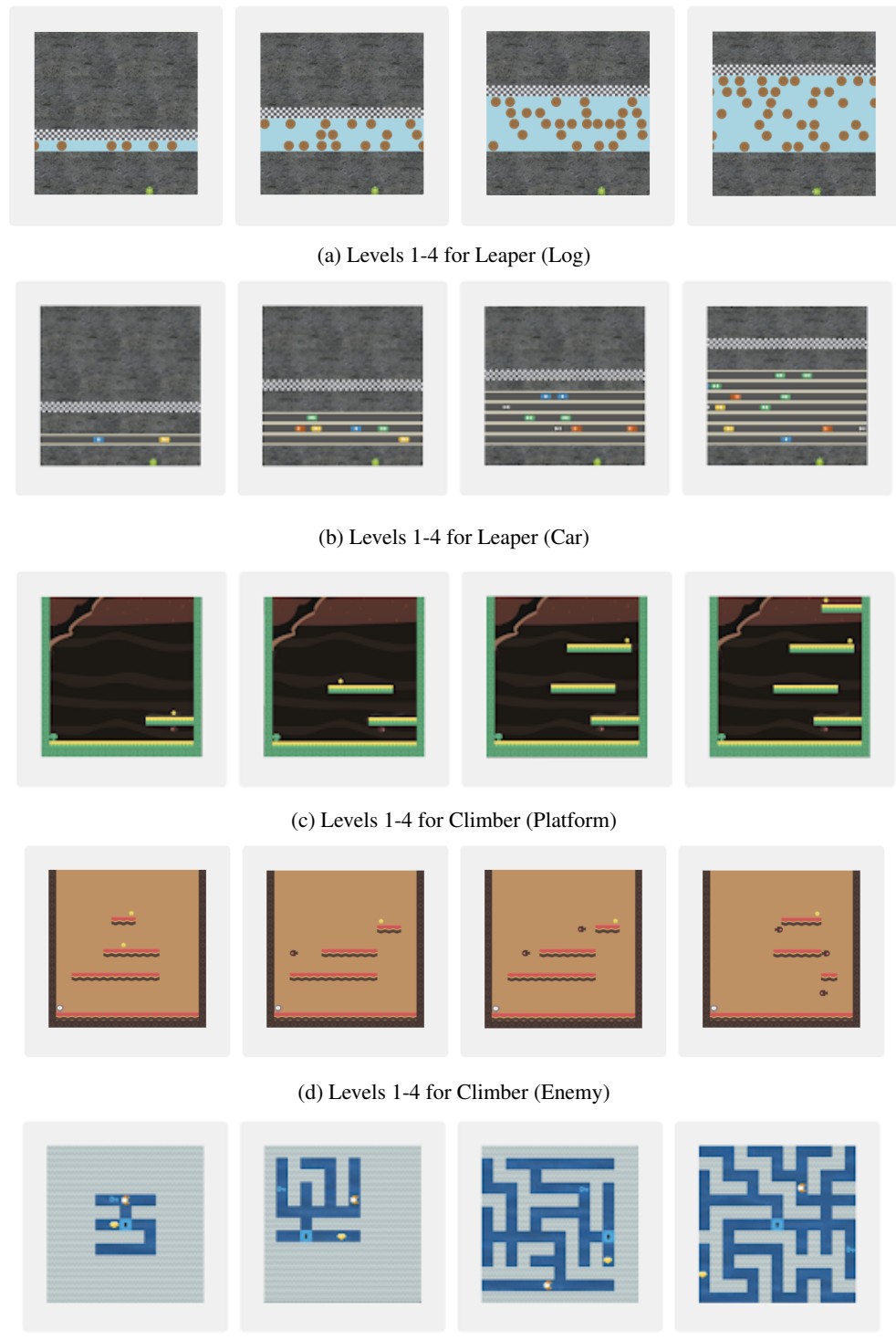

(a) Levels 1-4 for Leaper (Log)

(b) Levels 1-4 for Leaper (Car)

(c) Levels 1-4 for Climber (Platform)

(d) Levels 1-4 for Climber (Enemy)

(e) Levels 1-4 for Heist (Size)

Figure 7: We provide images of the Procgen game levels that are shown to participants. In each game, there are four levels varying in a single axis of difficulty. Level 3 is the target level.

## A.4 ADDITIONAL RESULTS ON CHILDREN'S CURRICULUM CHOICES

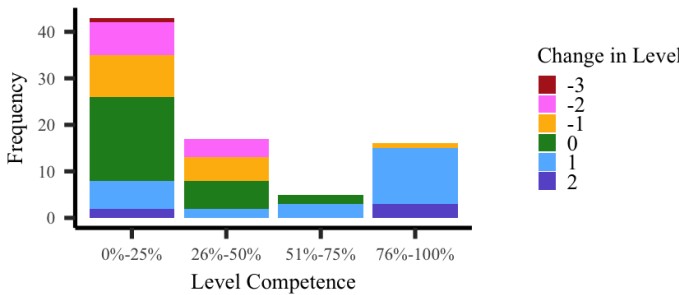

Figure 8: Level adjustments based on children's competence on the current Level with the inclusion of the extra challenging Level 4. Since there are a total of 4 difficulty levels in this case, the maximum possible absolute level change is 3. Whereas one participant made a level adjustment from Level 4 to Level 1, no participant made a level adjustment from Level 1 to Level 4.

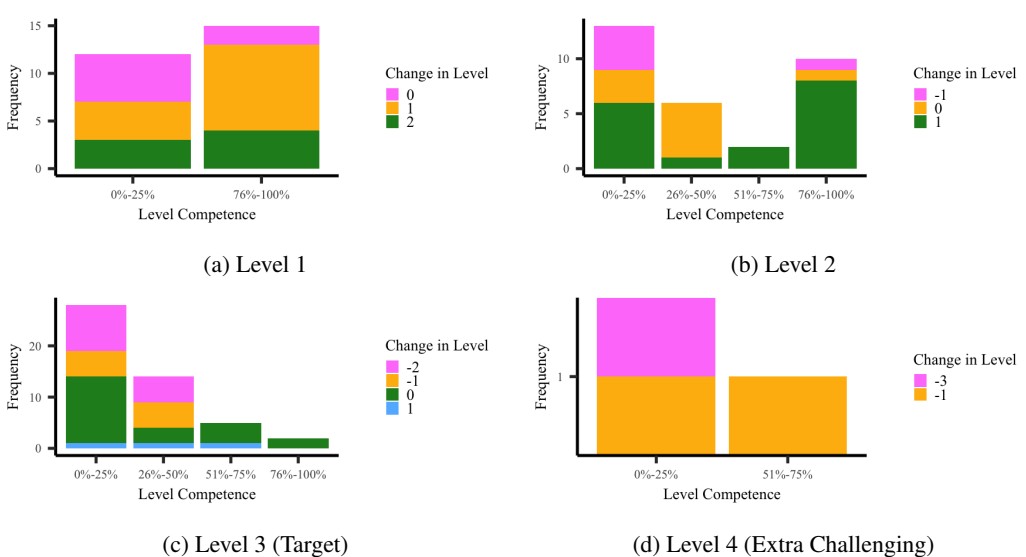

Figure 9: Children's level adjustments categorized by competence within each level. Note that only 2 participants selected Level 4 at any point of the curriculum.

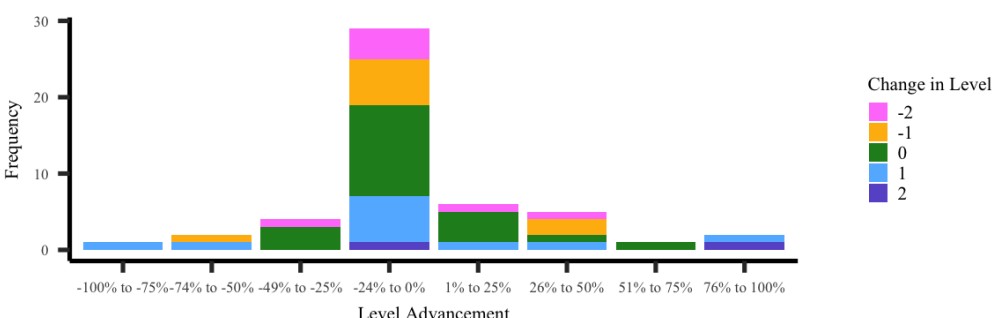

Figure 10: There was no significant correlation between children's advancement in the current level and their subsequent level adjustment.

A.5    ADDITIONAL BASELINES FOR RL EXPERIMENTS

This appendix details three additional baseline experiments that were conducted for the RL experiments. One involved randomly selecting levels, whereas the other two investigated a stochastic curriculum.

**Random level baseline**    In this experiment, we evaluate a baseline using a random curriculum: where the level is randomly selected from the possible distribution of levels. We conduct this experiment six times using only extrinsic rewards. We consistently saw poor learning performance, and the reward obtained on the target level was generally zero. This baseline is quite challenging as it is difficult to obtain a consistent learning signal from the extrinsic reward alone.

**Stochastic curriculum: selection of random levels**    In this experiment, 30% of the time, levels are chosen to be selected from the distribution of possible levels, from 1 water lane to 5 water lanes. In the remaining 70%, levels are determined based on the agent's current progress in the curriculum (as in Secs. 5.3 and 5.4). This experiment was conducted six times, and the agent was only trained using the extrinsic reward. In general, this strategy performs poorly. Four trials experienced training divergence. In only one trial was training still active at the end of the trial (getting to 3.0 water lanes). The average progression of the agent was 2.0313 water lanes, significantly less than in Sec. 5.3.

**Stochastic curriculum: selection of previously solved levels**    This experiment is similar to the previous stochastic curriculum experiment, except that instead of selecting from any level, previously solved levels are chosen. This experiment is conducted six times with only the extrinsic reward being used. None of the six trials experienced catastrophic forgetting. However, the agent does not progress through the curriculum as quickly as with an auxiliary reward (Sec. 5.4). The average progression in the curriculum was 2.9583 water lanes. Of the six trials, none could solve the target level, and only one trial reached 4 water lanes (whereas all six trials in Sec. 5.4 reached at least 4 water lanes, and five trials solved the target level). Therefore, learning and leveraging level competence as an auxiliary reward appears promising as not only a way to prevent forgetting, but also to advance the curriculum.

