# A    APPENDIX

## A.1    PROCGEN ENVIRONMENTS

There are 12 different Procgen environments. Figure 8 taken from Cobbe et al. (2020) showing all available Procgen environments.

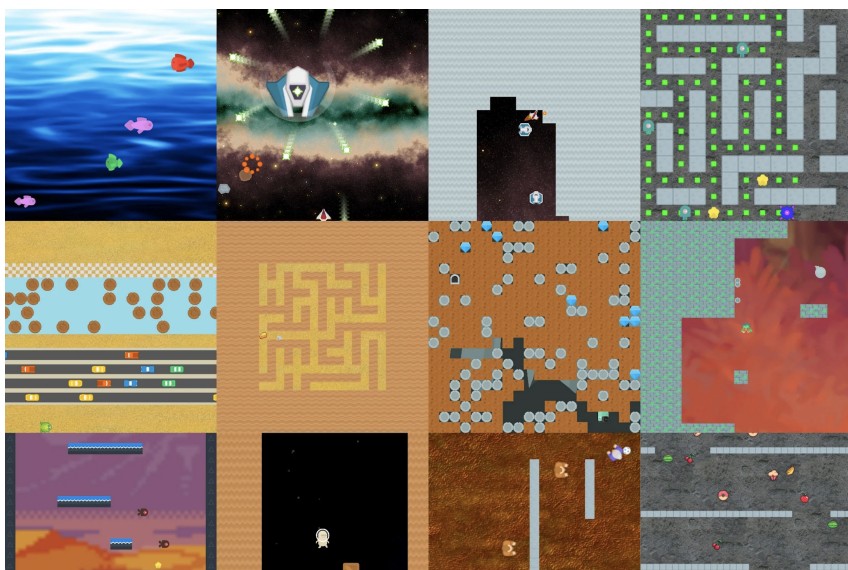

Figure 8: An illustration of all Procgen environments

## A.2    OUR ADAPTED PROCGEN ENVORIMENTS

| Game | Goal |
|---|---|
| Leaper Log | Cross the finish line and avoid going in the water as log lanes increase. |
| Leaper Car | Cross the finish line and avoid getting hit by a car as car lanes increase. |
| Climber Enemy | Reach the jewel and avoid enemies as number of enemies increase. |
| Climber Platform | Reach the jewel and avoid a single enemy as number of platforms increase. |
| Heist Size | Reach the final jewel by getting key and unlocking a lock as size of maze increases. |
| Heist Key | Reach the final jewel by getting key and unlocking locks as number of locks and keys increase. |

Table 2: We provide the goals for each of the Procgen games. Participants were never told the rules of the game and had to learn how to win the game through their own learning.

## A.3 PROCGEN LEVELS

We provide the levels that were shown to participants in the different Procgen games. There were four levels in varying difficulty, with Level 3 being the goal level.

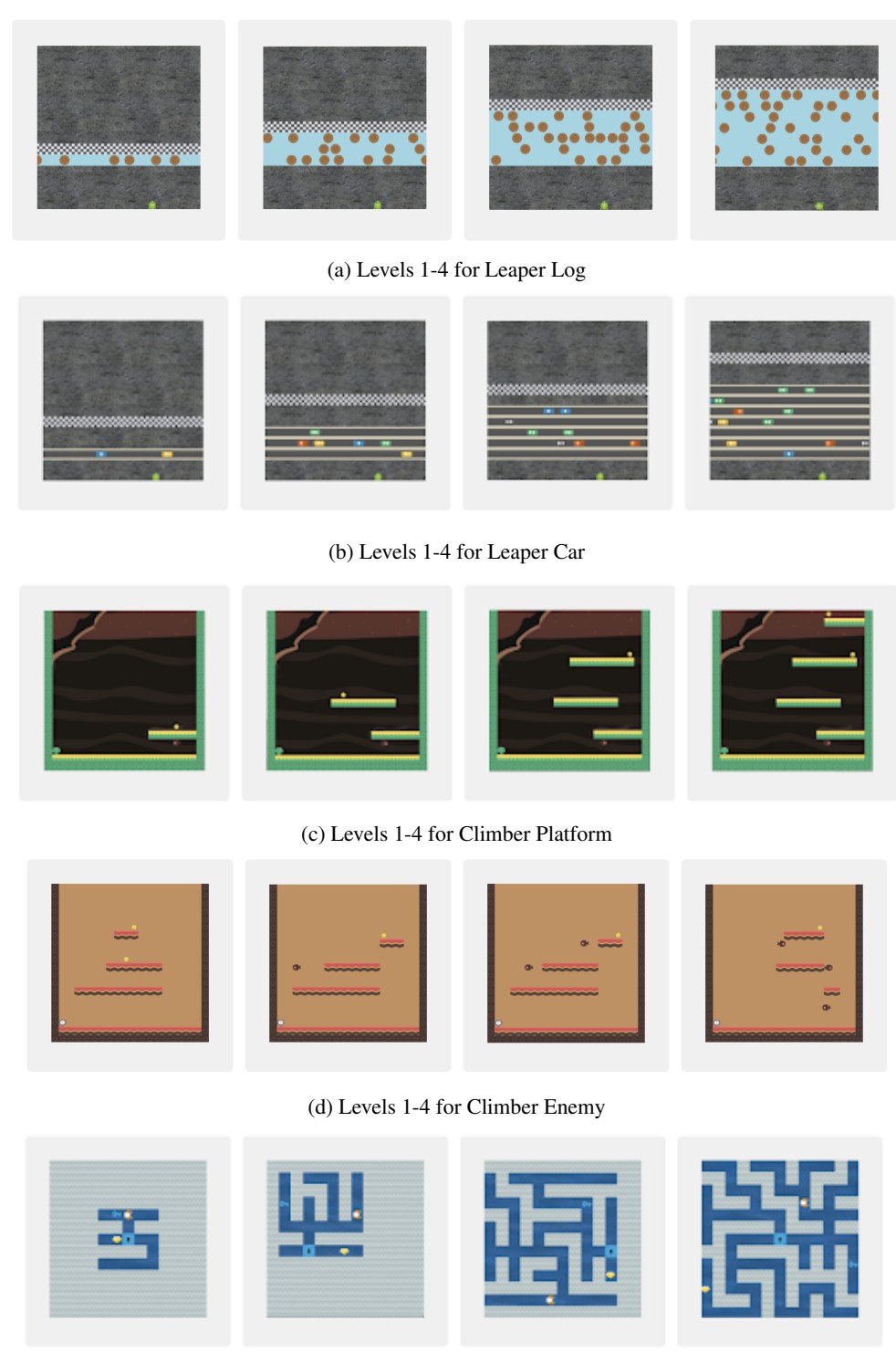

(a) Levels 1-4 for Leaper Log

(b) Levels 1-4 for Leaper Car

(c) Levels 1-4 for Climber Platform

(d) Levels 1-4 for Climber Enemy

(e) Levels 1-4 for Heist Size

A.4   LEVEL ADJUSTMENTS BASED ON CHILDREN'S PERCENTAGE OF PROGRESS ON THE
      CURRENT LEVEL WITH THE INCLUSION OF THE EXTRA CHALLENGING LEVEL 4

This figure shows the level adjustments children made when "irrational" selections of a level that
is challenging beyond the target level are included. Since there is a total of 4 difficulty levels in
this case, the maximum possible absolute level change is 3. Whereas one participant made a level
adjustment from level 4 to level 1, no participant made a level adjustment from level 1 to level 4.

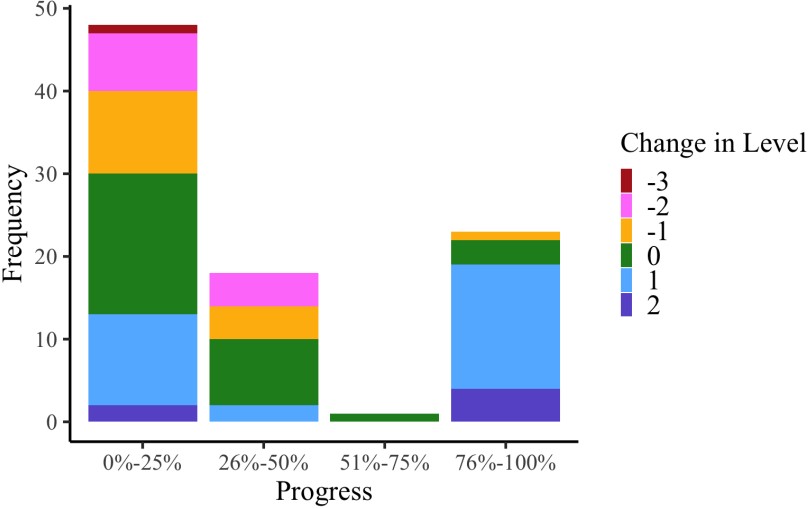

Figure 10: Level Adjustments Based on Children's Percentage of Progress on the Current Level with
the Inclusion of the Extra Challenging Level 4

A.5    CHILDREN'S LEVEL ADJUSTMENTS INDIVIDUALLY ORGANIZED BY PROGRESS ON
        LEVELS 1, 2, 3, 4

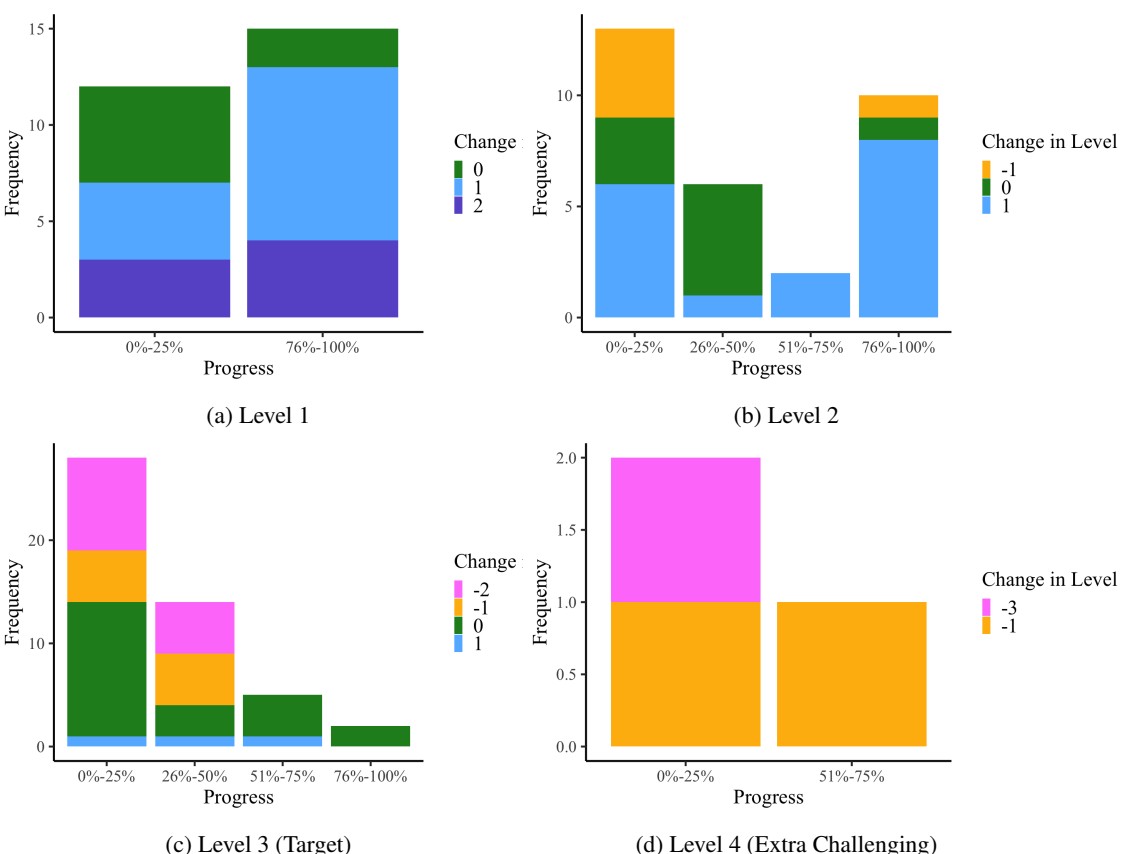

(a) Level 1

(b) Level 2

(c) Level 3 (Target)

(d) Level 4 (Extra Challenging)

Figure 11: Children made level adjustments based on progress within each level. Note that only 2 participants selected Level 4 at any point of the curriculum.