# OpenReview forum: "From Child's Play to AI: Insights into Automated Causal Curriculum Learning"
_ICLR.cc/2024/Conference — Submitted to ICLR 2024_

### Official Review · Reviewer_FUEG · 2023-10-16

**Soundness:** 2 fair
**Presentation:** 3 good
**Contribution:** 1 poor
**Rating:** 3
**Confidence:** 4

**Summary:**

This paper studies human and RL curriculum learning in a set of procedurally generated games.

**Strengths:**

Parallel studies of human and machine learning is a very exciting line of work that will help make progress in both disciplines. I think the paper is well motivated and clearly written.

**Weaknesses:**

Level progress is a confusing term for curriculum learning researchers. “Learning progress” or “competence progress” is a standard proposition of intrinsic motivation to guide curricula, it refers to the derivative of the competence measure with respect to time: how much the competence increases or decreases. Here you use the term “progress” but it refers to what people call “competence,” a measure of success, a score. For instance, if you play the same level 10 times and reach the same score each time, there is no progress at all across-episodes, the progress is only intra-episode. I know that level progress describes what you expect it to describe in the language of video games, but it’s confusing for researchers in this field: I thought you meant learning progress for the first 4-5 pages. Could you consider using another term? Competence is the standard term I think, score or performance could be good as well.

The hypothesis supported here is not very clearly stated. I can see several alternative hypotheses that could explain the data:
* Children have some form of heuristic curriculum strategy that would look a bit like this: if i’m very good, i switch to a harder task, if i intermediate I stay there, if i’m bad i either stay or try something easier.
* Children have a causal understanding of the world and maximize their causal learning
* Children have an intermediate difficulty bias (~Florensa’s paper): they keep doing the task for which they have intermediate competence / performance.
* Children have a bias towards learning progress: they can estimate expected learning progress in a model-based way and select the task that maximizes it. When a game is solved there is no further learning progress so they switch up, when they perform poorly they might expect more progress soon and stay, or not expect progress and recalibrate their estimations, which leads them to switch down.

I feel like the paper is arguing for the second interpretation, although it’s not stated clearly. I don’t see anything in the paper that would allow us to argue for one more than the others?

An interesting experiment could be to include levels from games different from the target game level. If children optimize for learning progress only, then we should see children select easy levels of the non-target games as well. Instead, we would probably see children almost never select the irrelevant game, which argues for a combination of intrinsic (LP-like) and extrinsic (going for the target level) curriculum.

RL experiments:
* The switch threshold appears to be an important hyperparameter, it would be interesting to see how it affects the results: .9 seems high given that children seem to switch up around 75%.
* I’m not 100% sure I understand the manual curriculum:
* why is there 16 parallel tasks?
* what does it mean to increment the difficulty by 1/16? I thought the difficulty was the number of lanes (1–5)? if the agent passes 0.9 in the current level (eg 1 lane), I expect the 16 tasks to become 2-lane tasks and the agent to be trained on these? Here it sounds like one of the 16 tasks becomes a lane-2 task, but then how is compute the score metric now? It it computed as the average level progress over the 16 taks (15 1-lane and 1 2-lane)?
* You use an on-policy algorithm that does not leverage a replay buffer. This means that old data from easier levels are thrown away. Using an off-policy algorithm (eg DDPG) would reuse past data, which may mitigate the catastrophic forgetting problem?

Catastrophic forgetting for curriculum learning is a known problem, which is why all curriculum approaches perform stochastic selection: they do not switch from one level to the other but sample all levels with varying probabilities that are a function of the intrinsic motivation. Eg selecting the current best level with probability .7 and the rest with .3 / n_other_levels may fix the issue, see Jiang’s paper and Colas 2019 for examples.

Studies of curriculum algorithms always include the presentation of the random baseline selecting level uniformly. This baseline does not suffer from distribution shifts.

I’m not sure how the intrinsic reward is used here. As far as I understand, it seems that the curriculum (level selection) is the same as before but that PPO now uses what the so-called intrinsic reward in addition, right?
If so, this is a problem. Intrinsic motivations must be agnostic of the goal (this is what intrinsic means). They can be either state-based: assigning an intrinsic reward to states; or goal-based: assigning an intrinsic reward to goals / tasks.
What you propose is to assign a reward to a state, but this reward is extrinsic, it measures performance in the task. Practically, you’ve just replaced a sparse reward with a dense reward, no intrinsic motivation here.
What curriculum people do in that situation usually is to use that competence measure to guide the level-selection (goal-based reward): the level selected could be the one where there is the most progress (score now vs score before, not the level progress), or the one where the competence (level progress) is intermediate. These rewards are extrinsic because they are not tied to particular levels: the level-selector only cares about learning progress or intermediate difficulty, not about any level in particular. This would involve using a bandit algorithm to explore levels and maximize that score.

The idea of a causal curriculum is interesting, but the papers cited when discussing this topic do not engage with any form of causality: eg Florensa 2018, Sukhbaatar 2017, Bengio 2009.

A good way of discussing curriculum approaches is by making explicit the distal objective of the curriculum (maximizing performance on a set of target tasks), describing the proximal objective optimized by the approach (which usually includes forms of intrinsic motivation), and discussing what is the control parameter (the part of the MDP that is varied to maximize the proximal objective), see framework from Portelas et al 2020. Proximal objectives include: novelty, intermediate difficulty, learning progress, things that can be varied include: state space, transition space, goals, rewards, etc. The current review is not very clearly structured and omits large chunks of the field (eg learning progress maximizing methods). These are probably the closest to the one proposed in this paper so they should be mentioned. In particular, this paper proposes a curriculum over tasks (variation of state space, and transition function) where the proximal objective is an intermediate difficulty (intermediate level progress).

What I would need to update my score:
* Further details and explanations for the points raised above
* Better account of existing curriculum approaches
* Explicit statement about the hypothesis proposed here + discussion about how the result support that hypothesis, and may or may not bring sufficient evidence to separate the different hypotheses i listed above
* I may have understood it wrong, but it seems that the authors completely missed the mark on the curriculum RL implementations: missing random baseline, deterministic level sampling instead of stochastic ones that's common in the field and, more importantly, the reward used is extrinsic and not intrinsic. It should be a goal-based reward and not a state-based reward. Again I may have understood it wrong so i'm open to discussions.

I believe that this setup is interesting and the scientific goal is good. I'm looking forward to future improved versions of this paper.


Minor comments:
* Vygotsky’s citation is broken (missing y, no year).

**Questions:**

see above

---

> ### Author Response · Authors · 2023-11-19
> **Response to Reviewer FUEG (1/2)**
>
> We thank you for your detailed feedback and suggestions. Your suggestions on the writing and experiments helped us to improve our paper, we hope to address your questions and concerns here.
>
> **Regarding the terminology of ‘level progress’.**
>
> We very much appreciate your  suggestion to better define level progress and other key terms for our paper. We have revised the terminology from ‘level progress’ to ‘level competence’, to reflect that this is not a derivative, but rather the current level of success the agent/ human achieves on the level. We have also written up more precisely defined definitions (see General Comments above).
>
> **Level Competence**: An indicator of success in an episode of game play on a specific level, reaching 100% when the specific level is successfully solved in that episode. (Note: This is the term formerly referred to as 'level progress.')
>
> **Global Competence**: A measure of success in an episode of game play with respect to the target level, reaching 100% if the target level is successfully solved in that episode.
>
> **Level Advancement**: Difference between current level competence and initial level competence within a particular level in the curriculum; it is a measure of how much competence increases or decreases in the same level
>
> **Global Advancement**:  Difference between current global competence and initial global competence across levels in the curriculum; it is a measure of how much competence increases or decreases with respect to achieving the target level
> Auxiliary reward: A scaled level competence
>
> **Regarding the alternative hypotheses that could explain the data.**
>
> We very much appreciate the suggestion to make our hypothesis more explicit, and we appreciate the reviewer for diligently differentiating between the hypotheses. The reviewer is correct in their understanding of what our proposed hypothesis is: Children have a causal understanding of the world and maximize their causal learning for a given goal (i.e., solving the target level). We appreciate the reviewer’s suggestion “to include levels from games different from the target game level.” We are currently collecting data for a follow-up study that addresses this question. We have a version of this study where we add an irrelevant “non-causal” axis, e.g., changing the color shades, and easy levels from other games. We will mention this follow-up study in the work. As an additional response to the reviewer’s question about examples of when children almost never select the irrelevant game, it can also be seen with our addition of the extra difficult level 4 (with the target being level 3) in the game design. Solving level 4 can be argued as irrelevant because it takes more unnecessary effort and does not help children achieve the extrinsic goal and win a sticker. We find that the majority of children never select the irrelevant, overly-difficult level 4 for building their curriculum to solve the target level.
>
> We also ran additional exploratory analyses to address your third and fourth alternative hypotheses:
> On intermediate difficulty bias: because the easier levels are direct subsets of harder levels in the games Leaper and Climber, we can calculate competence on easier levels as a percentage of competence in the target level to get global competence for each episode. We can then also determine which level children should objectively choose next by comparing their global competence with the percentage of the target level that each easier level represents. Here, we found that half of the time children were making objective choices, but another half of the time children chose harder levels than what they should objectively choose, suggesting that they have an intermediate to high difficulty bias that is likely also driven by the desire to causally solve the difficult target level
> On bias towards learning progress: we ran further analyses and found that children do not have a bias towards learning progress or advancement: our child participants neither used global advancement nor level advancement (as defined in our definitions in the General Comments) to guide their choice of next level to play, suggesting that they are not driven by how much progress they have made, but more by how well they are currently performing to move towards the goal (as demonstrated by our statistics on local competence guiding next level change)
> We will revise our paper to clearly state our hypothesis and include statistics of these additional findings.
>
> **Regarding the broken citation.**
>
> We fixed the broken Vygotsky citation, thank you for letting us know.

---

> > ### Comment · Reviewer_FUEG · 2023-11-20
> > **answer**
> >
> > Thanks for the detailed answer.
> >
> > I'm happy that the authors considered redefining terms, I think it's much clearer this way.
> >
> > About the hypothesis: Children have a causal understanding of the world and maximize their causal learning for a given goal (i.e., solving the target level).
> >
> > I see two ways of understanding this: are we talking about a causal understanding of the rules and physics of these games, that playing easy labels will help them causally learn those OR is the causal understanding that practicing on easier levels will cause improvement on target levels? I first thought the first was meant but now think the second is.
> >
> > I'm not sure I understand why changing the color shades would be a non-causal axis? Sounds like color is not causally related to anything to if the children were indeed to optimize for learning to improve on the target level they should sample these levels with different colors as well?
> >
> > I guess if you want to distinguish the different hypotheses, you might fit each of the models on part of the children and see how these model predict the level choices of the rest of the children?
> > eg. fit the intermediate difficulty thresholds on part of the children + then use that to predict the other's choices, another main parameter here is the window length: how many episode do you average to compute the given competence? just the last one or several?
> > what would be the predictions of your causal model? when should a causal learner pick each of the levels?
> >
> > Note that learning progress approaches use a local measure of learning progress, not progress since the start (same window length parameters, eg was i better in the last 3 episodes that i was in the previous 3?)

---

> ### Author Response · Authors · 2023-11-19
> **Response to Reviewer FUEG (2/2)**
>
> **Regarding additional baselines**
>
> We thank the reviewer for their suggestion on baselines to run. Please see the general comments for the baselines we are currently running and will report back on.
>
> We agree that the switch threshold is an important hyperparameter. We are also interested in seeing the results and will report back what we learn.
>
> To clarify the manual curriculum, there are 16 parallel tasks so that the agent can collect data more efficiently by interacting in parallel across 16 tasks. When we increment the difficulty by 1/16, this means that, of the 16 parallel tasks, one of the tasks has the difficulty incremented (and the others do not change). So, the progression would be 1) 16 1-lanes, 2) 15 1-lanes and 1 2-lane, 3) 14 1-lanes and 2 2-lanes, etc. We did originally experiment with switching all 16 tasks by one difficulty, but we found that this posed a sizable distribution shift to the agent that caused training divergence at much easier difficulties than if the difficulty was instead incremented by 1/16. However, we did notice as in Section 5.3 that training divergence does eventually occur. In Section 5.4, the level progress used in the auxiliary reward is calculated at the termination of an episode and then added to the extrinsic reward. For cases where we present average episode level progress (e.g., Figure 6b), this is calculated by taking the mean level progress over all finished episodes between policy updates.
>
> We thank the reviewer for the suggestion for trying an off-policy algorithm that uses a replay buffer. In the baselines that we just ran, we found that stochastically sampling previously solved levels indeed mitigates the catastrophic forgetting issue, however, progress made within the curriculum / towards solving the goal level is less than what we found in Section 5.4.

---

> > ### Comment · Reviewer_FUEG · 2023-11-20
> > **answer**
> >
> > I'm still unclear about the intrinsic reward.
> >
> > As I pointed out, it's not really intrinsic so it was renamed auxiliary reward. Am I right to understand that it rewards each step proportionally to progress in the current level?
> >
> > If so, this sounds unrelated to curriculum learning questions, it's just a way of doing reward shaping: having a dense rewards obviously outperforms sparse rewards alone, and it has nothing to do with level choices.
> >
> > So what is the curriculum part (level selection policy) of the suggested algorithm?
> >
> > Defining an RL algorithm on this tasks requires two things: a reward function, a level selection policy.
> >
> > * Switching a sparse reward for a dense one is obviously going to help a lot, especially since the dense reward is not deceptive in this case. This is reward shaping, not curriculum learning, and I'm not sure I see the relationship between this change and the human experiments.
> > * the human experiments seem to argue that children have a grasp of which levels are easier than others and have an intuitive sense of 'i should train on easier levels first': this is about level selection policy. Here you can compare different ones: maximizing intermediate difficulty, LP, random, hand-defined heuristic curriculum (if score>X, then switch to harder). What is exactly the new proposition made in this paper for the level selection policy? Each of these should be implemented with the same reward functions (either only dense, or only sparse, or both, but constant across level selection policies).

---

> ### Author Response · Authors · 2023-11-22
> **Response to additional comments for interpreting child data**
>
> We thank the reviewer for their time to follow up with additional comments for interpreting the child data.
>
> We are interested in both kinds of causal understanding (rules and physics of the game + practicing on easier levels causes improvement on target levels) and have ongoing experiments probing both ideas.
>
> We also informally looked at children’s open responses and found that they often referred to both the causal structure of the rules of the game and the causal structure of how solving an easier level could then enable them to solve a harder level of the same game.
>
> In our games, the causally relevant axis to help you achieve ultimate success at the goal level is difficulty. A child or an agent learns the game as a causal curriculum by choosing how to move along the difficulty axis. We consider changing the color shades to be a non-causal axis because there is no causal relationship between color shade and winning the game. Color does not have any significant causal power or insight in the game for achieving the target level.
>
> In our experiment setup, each child is subject to only a maximum of 10 rounds of automated curriculum learning; in every round they have the choice to select any of the game levels. Many children even solved the target level and terminated in less than 10 rounds (the average total number of rounds played is 7 across participants, with the minimum being 3). Thus, we do not have enough statistical power at the moment to adopt existing learning progress approaches (e.g., comparison between last 3 rounds and previous 3 rounds within a level) in our measure of progress or advancement. To circumvent this, we calculated level competence as just the competence in the active episode, and level advancement as the difference between the most recent episode and the initial episode of the same level.
>
> While we only have 22 children and are underpowered to do model fitting and model predictions, we have built linear mixed effects models to compare various hypotheses, and found that our data can be best explained by the idea that “children have a causal understanding of the world and use their level competence to guide their causal curriculum learning for a given goal (i.e., solving the target level)” (this model has the lowest AIC and BIC). After further examination of the data, intermediate difficulty bias and level advancement (not computed as the classic learning progress due to limited data per individual participant) so far did not explain much of child behavior - a notable portion of children seemed to remain at the same level or move to higher levels even when they had very low level competence or negative level advancement because they were motivated to achieve the extrinsic goal of solving the target level, but we have yet to collect more data to confirm this idea.
>
> Once we have collected enough data, we will definitely try adopting your recommendations on fitting different models with some portion of child data and using them to predict other children!

---

> ### Author Response · Authors · 2023-11-22
> **Response to additional comments for RL experiments**
>
> We thank the reviewer for their time to follow up with additional comments for the RL experiments.
>
> Regarding the auxiliary reward, at the end of each episode, the agent receives a reward that is the sum of 1) an extrinsic reward (which, in this case, only occurs if the level is solved) and 2) the current advancement of the agent within the level. The agent only receives these rewards at the end of an episode, not at each step. We agree that the consequence of providing such an auxiliary reward is that it “densifies” the rewards.
>
> Although it is not directly curriculum learning, we see that it impacts curriculum learning results. Having such a reward helps prevent forgetting which arises due to the distribution shift induced by switching levels. It may not be the only way to prevent forgetting, but having the auxiliary reward appears promising to both prevent forgetting and advance the curriculum. In the baseline experiment we recently conducted using a stochastic curriculum with sampling previously solved levels, we see that forgetting was also avoided, but the agent made less progress within the curriculum than with auxiliary rewards.
>
> The relationship between the auxiliary reward in the RL experiments and the human experiments is that we observed that the children’s competence within a level improved without necessarily solving the level. The analogy for the RL agent is that if there was an additional reward besides the sparse extrinsic reward that could help with learning, the agent could potentially learn more effectively than with extrinsic rewards alone.
>
> At the present time, our RL experiments were conducted using a hand-specified curriculum. In future work, we are interested in experimenting with different level selection policies. However, our results do show that, at least for this task with sparse rewards, that other algorithmic capabilities besides level selection (e.g., capabilities for preventing forgetting that arises when switching levels) may be needed for effective curriculum learning.

---

> > ### Comment · Reviewer_FUEG · 2023-11-22
> >
> > I still disagree with the treatment of the 'auxiliary reward'. It's purely a dense version of the extrinsic reward, it is an extrinsic reward: it's assessing progress towards the resolution of the task. It helps learning for that reason (a known result), and I'm not sure in which sense it can be said to 'help curriculum learning'.
> >
> > Both curriculum learning (level selection) and reward shaping impact learning (task success): the dense extrinsic reward changes reward shaping, not curriculum learning, which in turns helps learning.
> >
> > The choice of providing the dense extrinsic reward only at the end of the episode and not at every step does not sound very natural to me, is there a specific reason for doing so? The RL algorithm will need to backtrack this reward to the beginning of the episode via value iteration and that sounds like a harder job then just assigning dense rewards at every step.
> >
> > "The relationship between the auxiliary reward in the RL experiments and the human experiments is that we observed that the children’s competence within a level improved without necessarily solving the level."  --> the key difference here is that children have world models and can thus anticipate future rewards they have not yet collected. Using a dense reward function slightly compensates for this by modeling 'intermediate' rewards.
> >
> > Overall I don't really see the point of the RL experiments: they only show that using a dense extrinsic reward works better than a sparse one (known result), and do not provide any extra insight into the particular curriculum strategy (level selection) that children follow beyond what the human experiments tell us.
> >
> > ---
> >
> > I'm also still unclear about the claim of 'causal learning'. If children actually understood the true causal structure of the task, couldn't they always train on the target task? I think I've seen written somewhere that harder tasks are a superset of easier tasks: so training on easier tasks is strictly equivalent to training on the beginning of harder ones? If they prefer training on easier tasks first, we can't only explain this by saying they exploit the causal structure of the task. There must be something else: maybe they like achieving intermediate rewards, eg they could assign reward 10 for the target, 7 for level 3, these would be 'internal extrinsic rewards', as the experiment design doesn't really reward them for easy level completion? Or it could be about intrinsic motivation (LP, intermediate difficulty). As such, the experiments do not really allow to claim that children are motivated by the target task only and exploit the causal structure of the problem. If so, some of them should simply always train the target task.
> >
> > Another hypotheses is that they have a 'folk model of causal learning' and have the intuition that easier level should be solved before harder ones?

---

### Official Review · Reviewer_MFeg · 2023-10-31

**Soundness:** 3 good
**Presentation:** 3 good
**Contribution:** 1 poor
**Rating:** 3
**Confidence:** 4

**Summary:**

This paper first exams the learning strategies of children in tackling challenging tasks. The study used Procgen environments with various difficulty levels to investigate how 5 to 7 years old children adapt to these challenges. The findings reveal that children use their progress through the levels as an intrinsic reward, motivating them to excel at easier levels before tackling harder ones.

Then this paper shows that RL agents that follow a similar approach, incorporating level progress as an intrinsic reward, show better stability and convergence during training than agents relying solely on extrinsic rewards. Curriculum designed in this way also boost the sample efficiency.

**Strengths:**

(1) The analysis on children playing Procgen games are enlightening and thorough.
(2) The idea of adopting the behavioral findings from human studies to solve ProcGen is inspiring.

**Weaknesses:**

(1) Level progress is not defined clearly at all through the whole paper leaving the other discussion built upon it shaky.
(2) From my understanding after reading the texts, level progress seems to be a rough measure of how far you have gone in this level. Putting aside how we can access such information, if we do, this will vary from environment to environment and it seems to be simply a denser supervision signal for each move the agent makes towards accomplishing the task. So, the proposed method is both inaccessible and not generalizable.
(3) In the experiment section, the authors didn't show training results with multiple random seeds but only single runs.

**Questions:**

Questions are related to the weaknesses section.
(1) What is the formal definition of level progress?
(2) How do you plan to access the level progress you defined?
(3) Could you do a parameter search and change the network structure with multiple random seeds to see if it's poor training causing the catastrophic forgetting? In another word, is it really because the sparse reward signals?

**Details Of Ethics Concerns:**

I am not sure if the authors have been authorized to do human subjects experiments as there are no such sections stating that in the paper. So, I would suggest a double check.

---

> ### Author Response · Authors · 2023-11-19
> **Response to Reviewer MFeg**
>
> We very much appreciate the reviewer’s time and feedback.
>
> **Regarding the terminology surrounding “level progress”**
>
> We would like to thank the reviewer for their suggestion to better define level progress and other key terms for our paper. We have introduced a definition table, which contains clearer explanations. These definitions are presented below. Level Competence (previously referred to as ‘level progress’): an indicator of success in an episode of playing a level, reaching 100% when the specific level is successfully solved in that episode.
>
> **Regarding authorization to do human subject experiments**
>
> We thank the reviewer for checking on authorization to do human subject experiments. We have IRB approval to conduct human subject experiments and will include a sentence in our experimental design that the experiments have been approved by our university’s IRB.
>
> **Regarding inaccessibility and generalizability of level progress**
>
> We would appreciate it if the reviewer could clarify their concerns regarding the inaccessibility and generalizability of level progress. We agree that, in our current formulation, the use of level progress as an auxiliary reward does have the effect of providing a denser supervision signal. To the question of how we plan to access level progress, we believe the work of Bruce 2023 “Learning About Progress From Experts” (as mentioned by reviewer Rafo) could be extended to our domain.
>
> **Regarding the number of random seeds**
>
> As for the RL experiments, each of the experiments in Section 5.3 and 5.4 are run over six trials each, where each trial has a unique random seed. Figure 4 and Figure 6 are from one representative trial within the six trials.
> For Section 5.3, we consistently saw catastrophic forgetting across random seeds. The hyperparameters used were those from Jiang 2021 “Prioritized Level Replay”.

---

> ### Comment · Reviewer_MFeg · 2023-11-21
>
> Thank you for your response. However, I am still confused about some important aspects.
>
> 1. It is clear that 100% of a level progress means the completion of the level. But how do you define the progress formally when it is sth like 8%, 67%, 93%? Does there exist a unified notation of such progress for all the environments? At least in the suite of Atari games, I personally find such a level progress hard to define.
>
> 2. How did you calculate the level progress in your current experiments? Is it manually assigned?
>
> 3. If the experiments were conducted in multiple random trials, I am expecting to see the confidence intervals (e.g., a shaded area around the curve) instead of a single curve. And for fig.4 and fig.6, I would suggest the authors present the full results instead of hand picking a "representative trial", which seems not to be a good practice. You may refer to Agarwal et al. [1] for a recommend way of reporting results in RL.
>
> [1] Agarwal, Rishabh, et al. "Deep reinforcement learning at the edge of the statistical precipice." Advances in neural information processing systems 34 (2021): 29304-29320.

---

> > ### Author Response · Authors · 2023-11-22
> > **Response to comments on notation for level progress and figures**
> >
> > We thank the reviewer for their time to follow up with additional comments for calculating level progress and the RL training figures.
> >
> > In our current experiments, level progress (now “level competence” to avoid confusion) was not manually assigned. In all our games, we developed clear and objective code within the Procgen games to compute progress in an active episode. Here is a breakdown of what it means in each of the 3 games:
> > -In Leaper, there is a number of vertical lanes that an agent has to cross in order to reach the finishing line; here progress in an episode is calculated as the vertical lane that the agent has reached, normalized by the total number of lanes between the start and the finish line
> > -In Climber, there is a number of platforms that an agent has to hop onto in order to reach the final star reward; here progress in an episode is calculated as the platform that the agent has reached, normalized by the total number of platforms between the start and the finish line
> > -In Heist, there are a number of objectives in the game including gathering keys, unlocking locks, then gathering the jewel; here progress in an episode is computed as how many objectives the agent has achieved, normalized by the total number of objectives
> > note: there are 2*L + 1 objectives in Heist, where L is the number of locks. So, a game with 3 locks would have 7 objectives (collect 1st key, unlock 1st lock, collect 2nd key, ... , collect jewel); collecting the 1st key would lead to 14.2857% progress, unlocking the 1st lock would lead to 28.5714% progress, and so forth.
> >
> > Figure 4 and Figure 6 were chosen as highlighting one trial as they were emblematic of the six trials that were conducted for each of these experiments. We felt this would be less visually distracting, as the six traces would either diverge at different points (as in Fig 4 / Sec 5.3) or finish training at different points due to solving the goal level (as in Fig 6 / Sec 5.4). This is different from training RL agents for a certain, fixed number of training frames, which would be amenable to showing all of the trials on one visualization. In our case, our training will conclude early if the agent can solve the goal level.

---

> > > ### Comment · Reviewer_MFeg · 2023-11-23
> > >
> > > First, I still think the contribution of the paper is minor since the proposed level progress is basically an extrinsic reward that even requires environment specific engineering, which greatly restricts its application range. And this extrinsic rewards, which as also pointed out by other reviewers, doesn’t have a strong connection to the curriculum learning, nor it is new.
> > >
> > > Second, the authors seem to be oblivion to the rich, rigorous literature of causal inference. Without defining any causal models throughout the paper, somehow the paper title is claiming about “causal learning”.
> > >
> > > Thirdly, the experiment results reporting is biased. I still cannot be convinced that one should hand pick runs for better presentation.
> > >
> > > Overall, I’ll keep the score unchanged.

---

### Official Review · Reviewer_tuxP · 2023-10-31

**Soundness:** 3 good
**Presentation:** 1 poor
**Contribution:** 2 fair
**Rating:** 3
**Confidence:** 3

**Summary:**

Based on the observation that children use their current level progress to determine the next level in their curriculum, this study proposes a curriculum approach for reinforcement learning agents. Using level progress as an intrinsic reward signal, similar to the way children do, the study argues for improved data efficiency and convergence in reinforcement learning agents.

**Strengths:**

- This study draws inspiration from observing real 5-7-year-old children, making it interesting as a human-inspired finding.
- The intrinsic reward method proposed in this study reduces catastrophic forgetting and improves convergence.

**Weaknesses:**

- The analysis of the behavior of 5-7-year-old children, obtained through observation, is intriguing, but the transition to rl-agent experiments is somewhat lacking. While the study claims to use an intrinsic reward, it seems to be primarily based on "how far the agent is into the task," leaving questions about whether this intrinsic reward function has more significant implications than the distinction between "sparse reward" and "dense reward" typically used in the general RL community.
- The explanation of the experimental results is somewhat vague, and the experimental environment appears to be limited.
- If the way 5-7-year-old children change their levels does not apply to RL agents, it suggests that the study has not fully integrated the observations from children's behavior into RL agents.
- There is no comparison with baselines or prior work.

**Questions:**

- In both Experiment 5.3 and 5.4, does the agent move to the next level when it achieve an average reward of 9 means (as explained in section 5.2)?
- Did the proposed method in 5.4 incorporate the way 5-7-year-old children change their levels?
- While it's mentioned that 5-7-year-old children use level progress as an intrinsic reward signal to decide which level to play to solve the most challenging level, does the proposed method in section 5.4 actually use this intrinsic reward function (2* lambda / 100) for transitioning between levels? (i.e. do they use it when they determine whether they change their level or not?, or just determine the level transition based on if(mean episode reward>9)? )
- How many seeds were used to obtain the experimental results?

---

> ### Author Response · Authors · 2023-11-19
> **Response to Reviewer tuxP**
>
> We very much appreciate the reviewer’s time and feedback.
>
> **Regarding clarifications on the experimental results and environments.**
>
> We appreciate the reviewer's suggestion regarding clarifications on our experimental results and the  experimental environments. Could we ask the reviewer to specify additional details or highlight specific areas of concern for us to address.  We are enthusiastic about enhancing the clarity of our explanations and offering a more comprehensive understanding of our experimental setup.
>
> **Regarding additional baselines.**
>
> We thank the reviewer for their suggestion on baselines to run. To clarify the experiments in Sections 5.3 and 5.4, the agent advances to the next level of the curriculum when the average reward obtained exceeds 9. The auxiliary reward mentioned in Section 5.4 is inspired by human curriculum learning, but the advancement of level is still just done when the average reward obtained exceeds 9. We are interested in running a baseline where the curriculum also changes based on lambda.
>
> **Regarding the number of random seeds for our RL experiments.**
>
> For the experiments in Sections 5.3 and 5.4, six trials were used for each experiment, where each trial has a unique random seed. The results shown in Figure 4 and Figure 6 are for one representative trial within the six trials.

---

> > ### Comment · Reviewer_tuxP · 2023-11-20
> >
> > Thank you for your response.
> >
> > If I understand correctly, in the 5.3 and 5.4 experiments, the agent moves to the next level when the average reward exceeds 9. Does this mean that the integration of insights gained from analyzing the behavior of 5-7-year-old children in this study only provides "how far the agent has progressed" as an intrinsic reward?
> >
> > If so, is there any difference other than changing "sparse reward" to "dense reward" that is commonly used in the RL community?
> >
> > Additionally, I understood that Figure 3 illustrates not how children construct intrinsic rewards but rather how they determine transitions between levels. Nevertheless, it is regrettable that this study did not integrate such inspiration into the movement between levels. My understanding of curriculum learning primarily focuses on the consideration of which level of the task to provide, rather than how to reward. Therefore, there remains a question about whether the proposed approach can be considered curriculum learning, given that it seems to focus more on how rewards are given.
> >
> > Thank you.

---

> > > ### Author Response · Authors · 2023-11-22
> > > **Response to additional questions regarding the RL experiments**
> > >
> > > We thank the reviewer for their time to follow up with additional questions regarding the experiments.
> > > Correct, in the RL experiments, we use a hand-designed curriculum that switches to a more difficult level when the average reward exceeds 9. (In 5.4, this threshold only looks at the average extrinsic reward, to make a fair comparison to 5.3 when switching levels).
> > > The insights from the children studies in how children can learn and increase their competence without necessarily solving the level inspired our study in 5.4 of seeing if, assuming a metric was available to assess level competence, using this metric as an auxiliary reward to improve learning. Indeed, we found this was the case: it prevented catastrophic forgetting. The effect of training using this auxiliary reward can indeed be seen as a way of “densifying” the reward signal. (Note that during the rebuttal, we changed the terminology of the “intrinsic reward” to “auxiliary reward” to better align with the literature in this area.)
> > > We also believe it would be useful in future work to integrate how children change levels (Figure 3) into the curriculum learning policy for the RL agent (how it changes levels). Nonetheless, we believe the insight of how children learn without necessarily solving the levels is important in the design of a curriculum learning agent. This is because curriculum learning involves intentionally inducing distribution shifts, so capabilities that mitigate the consequences of that (such as using an auxiliary reward) would be fruitful for effective curriculum learning.

---

### Official Review · Reviewer_Rafo · 2023-11-01

**Soundness:** 2 fair
**Presentation:** 4 excellent
**Contribution:** 2 fair
**Rating:** 5
**Confidence:** 3

**Summary:**

The authors study curriculum learning in reinforcement learning by conducting human experiments on complex sequential tasks. They discover that children successfully solve the final complex tasks by taking the correct curriculum arrangement according to the task progress. Motivated by this, the authors design a reinforcement learning experiment to verify this finding by setting progress level as an intrinsic reward. The experimental results indicate that the progress level intrinsic reward improves the curriculum learning performance.

**Strengths:**

1.	The authors conduct an interesting and informative human experiment.

2.	The paper is well-organized.

**Weaknesses:**

Please refer to the questions below.

**Questions:**

1.	I’m sorry that I’m not very familiar with the human experiments in cog-science. But I worry that the number of children (only 22 children) in this study is too small. How do you guarantee that the results obtained from such a small group of children is unbiased and trustable? Maybe please point us to some related works that also conducted human experiments and claimed that such a small experimental population would be enough.

2.	Another concern is about novelty. I admire the motivation from human experiments. However, utilizing task progress is not a novel idea in reinforcement learning [1], so I worry that the proposed method is more like a trick.

3.	There are many previous works taking advantage of various heuristics to design curriculars. I think it is necessary to compare your method with them. Would you please show us some comparison results with existing techniques?

[1] Bruce J, Anand A, Mazoure B, et al. Learning about progress from experts

---

> ### Author Response · Authors · 2023-11-19
> **Response to Reviewer Rafo (1/2)**
>
> We very much appreciate the reviewer’s time and feedback.
>
> **Regarding the number of children in the study.**
>
> We are collecting more data on children at the moment. There are developmental cognitive science papers that have published experiments with a sample size of about 20-30 each (e.g., see Experiment 1-3 in [3],  see Experiment 1 in [4]). (Children are much harder to recruit than adults, but they are important for understanding the bases of curriculum learning.)
>
> But more importantly, even though 22 children looks like a small sample size, each child does not only contribute to one data point. We are interested in examining children’s steps (or rounds of the game played) throughout their curriculum; within each step, we have details about which level they select, how well they perform on the specific step, etc. We currently have data for 159 steps that children made in their curricula and even when we factor in individual differences in our regression model, we are still able to find a statistically robust pattern between their level competence and their choice of next level.
>
> [1] Kushnir, T., Gopnik, A., Chernyak, N., Seiver, E., & Wellman, H. M. (2015). Developing intuitions about free will between ages four and six. Cognition, 138, 79-101.
> [2] Sobel, D. M., & Corriveau, K. H. (2010). Children monitor individuals’ expertise for word learning. Child development, 81(2), 669-679.
>
> **Regarding additional baseline experiments.**
>
> We appreciate the reviewer for pointing this out.  thank the reviewer for their suggestion on baselines to run.  We have indeed conducted additional experiments, and we report the results here. First, we have conducted a baseline where environments are chosen randomly from 1-5 water lanes, without the curriculum. This is done for six trials/random seeds with a limit of 10M frames. We find that the agent makes very little learning progress over 10M frames. The return for the goal level (5 water lanes) is generally zero over the entire 10M frames. Moreover, the agent does not obtain much training return, either. We hypothesize that this is a difficult exploration problem for the agent without a curriculum, and although the agent may eventually make progress, the timescale would far exceed the 10M frame budget we used for our experiments.
>
> Second, we conducted a baseline against Section 5.3 (curriculum without auxiliary reward) where, 30% of the time, previously solved levels are chosen based on where the agent currently is in the curriculum. For example, if the current curriculum progress is 3.5 water lanes, then levels would be chosen from [1, 2] water lanes 30% of the time. This is done for 6 trials/random seeds with a limit of 10M frames. We do find that this mitigates catastrophic forgetting, as training divergence did not happen in any of the trials. However, the agent makes less progress towards the curriculum than in Section 5.4 with the auxiliary reward. In Section 5.4, all six trials got to at least 4 water lanes, whereas in this baseline, only one trial got to 4 water lanes. The average number of water lanes across all trials was 2.958. Additionally, in this baseline, none of the trials were able to solve the goal level (unlike in Section 5.4, where 5 of the trials could solve the goal level). We find that, although stochastically selecting previously solved levels prevents catastrophic forgetting, it may not be an optimal approach for advancing the curriculum. We hypothesize that this occurs because 30% of the training budget is being spent to prevent agent forgetting, rather than learning how to solve the latest level in the curriculum. Therefore, it would be valuable to have an agent learn and utilize the auxiliary reward used in Section 5.4, as this both prevents catastrophic forgetting and advances the curriculum.
>
> Third, we conducted a baseline that was similar to the second one, except instead of sampling previously solved levels, all possible levels (from 1 to 5 water lanes) were sampled. Six trials/random seeds were used with a limit of 10M frames. This baseline generally does poorly. In 4 trials, the training divergence still occurred. In only one trial was the training still active; this trial reached 3.0 water lanes. We hypothesize that this baseline performs worse than sampling previously solved levels, because sampling more difficult levels generally leads to more difficult levels that may not be solvable. This yields sparser returns and induces the same training divergence behavior seen in Section 5.3.
>
> We hope this has addressed the reviewer’s concern, we would be happy to discuss/ conduct any additional experiments if the reviewer has additional experiments/ concerns in mind.

---

> ### Author Response · Authors · 2023-11-19
> **Response to Reviewer Rafo (2/2)**
>
> **Regarding related work by Bruce 2023.**
>
> We also thank the reviewer for bringing Bruce 2023 to our attention, as this work seems highly relevant. Indeed, the concept of task progress is not novel to our work. However, we note a difference between our setting and Bruce 2023 is that Bruce 2023 requires expert trajectories to learn their progress model. In our case, we do not have access to expert trajectories, so we can envision that our approach could introduce novelty by relaxing this assumption and thereby extend their method.

---

### Official Review · Reviewer_5JBB · 2023-11-04

**Soundness:** 3 good
**Presentation:** 4 excellent
**Contribution:** 4 excellent
**Rating:** 6
**Confidence:** 4

**Summary:**

The paper studies intrinsic learning in children and how they use ‘level progress’ information to modulate their understanding of goals and make progression towards task goals. This automated curriculum selection in children isn’t random (as shown with experiments) and helps them learn. Authors use the same principle to design RL agents to solve simple game playing tasks with varying levels and design hand-crafted curriculum and conduct experiments to understand how RL agents learn in the absence of extrinsic rewards and whether they are able to recover from failures. The baseline they develop isn’t able to achieve goals as difficulty levels increase. “Although for simpler environments it may be possible to recover, as the complexity is smaller and random actions may find the goal, for harder environments with multiple lanes, this divergence is unrecoverable.” From Fig 5b, Before training divergence (which happens with increasing levels of difficulty and catastrophic forgetting), the exact relationship of reward and level progress depends on the task complexity. The easiest task (1 water lane, dark blue) has the greatest slope, since changes in level progress yield relatively greater mean training reward. However, this pattern does not transfer as the complexity increases. With inspiration from how children use the level progress as a proxy for reward signal, authors conduct experiments using the similar level progress information as intrinsic reward signal. RL agents learn much better and can recover from catastrophic forgetting when using this intrinsic progress as a reward signal after every episode. Intrinsic rewards decrease as levels progress however they do not collapse and can increase (recover) in certain circumstances. Authors suggest understanding how to integrate these intrinsic level progression into RL agents from the world states (eg. From high dimensional images) will be crucial towards the path for more general learning agents.

**Strengths:**

- Design the complex experiments to understand how small children learn using simple games that contain difficulty levels and drawing inferences on what signals children utilize as intrinsic signal for achieving goals is a very good contribution of this study. Many ideas in the paper seem intuitive; however, these scientific experiments and evaluation to generate various hypothesis is commendable.
-  The paper is equally divided into a user study of children to understand how they use automated curriculum learning to solve tasks and then explore how RL agents can learn on same tasks using similar automated learning strategies. Authors show promising results and future directions of the work to highlight where the field should focus to utilize automated curriculum learning with RL agents and how to design these intrinsic rewards.

**Weaknesses:**

- Although very inspirational work, there are many questions around the user study that is not clear. How reliable are children responses? How was this controlled for and simplified so children could understand and respond appropriately?
- Asking 5 year olds multiple choice questions pertaining to the true causal rules of games needs more supportive evidence.
- These are small experiments - Children’s selection of an easier level after failing a level is spread over different levels (including 1/7 choosing a more difficult level). 42.1% of failed levels led to choosing an easier level, 37.4% of failed levels led to choosing the same level. - how reliable is this and it is not clear how the authors are utilizing any of this fine-grained signals in the intrinsic reward design process for RL agents.

**Questions:**

(please address the points under weakness section)
- RL experiments are done with only Leaper game. Are the results similar on other games?

Minor comments:
- Table 2 missing
- Section 3, 5th line - unclear - “we heightened the difficulty by increasing the number of platforms” - what’s a platform here?
- Fig 3 references fig 9 and fig 10 in appendix (not present)

---

> ### Author Response · Authors · 2023-11-19
> **Response to Reviewer 5JBB**
>
> We appreciate the reviewer for their positive and detailed feedback. We attempt to address the questions raised by the reviewer and would be more than happy to address any additional questions that the reviewer may have.
>
> **Regarding questions on the user study of children.**
> We agree that it is important to have a well controlled and understandable setup for children. In our experiment setup, we included an extra difficult level beyond the target level to validate that children were not just making random choices: we found that so far only 14% of children selected to play this extra difficult level, suggesting that most children understood what their goal was and did not attempt what was beyond that goal. Furthermore, we observe statistically significant evidence that children use their game level competence to guide their decision of which level to play next, again reflecting that they understood their task and were not making random responses.
> With regards to multiple choice questions, there are developmental studies studying children of similar age ranges that use multiple choice questions [1],[2]. We will include relevant citations in our paper. More importantly, we found that most children were able to respond correctly to the multiple choice question pertaining to the true causal rule of the game.
> [1] Kominsky, J. F., Gerstenberg, T., Pelz, M., Sheskin, M., Singmann, H., Schulz, L., & Keil, F. C. (2021). The trajectory of counterfactual simulation in development. Developmental Psychology, 57(2), 253.
> [2] Rafetseder, E., & Perner, J. (2018, April). Belief and counterfactuality: A teleological theory of belief attribution. Zeitschrift für Psychologie, 226 (2), 110-121. doi: 10.1027/ 2151-2604/a000327
>
> **Regarding additional RL experiments.**
> We appreciate the reviewer for pointing this out. We are running experiments  on other games in Procgen. We will try our best to complete the experiments within the time constraints, and we will report the results as soon as they are done."

---

### Author Response · Authors · 2023-11-19
**General comment to all reviewers (1/2) - definitions**

We thank all the reviewers for their time and detailed feedback on our paper. Feedback by the reviewers have been very helpful. We are also glad that the reviewers found our paper that  adopts the behavioral findings from human studies to solve RL tasks to be inspiring and interesting. We would like to address two points that have been raised, extending our response to all reviewers.

**Definitions**:
Some reviewers have highlighted the lack of clarity surrounding the term 'level progress,' expressing concerns about potential confusion. In response, we have included a dedicated section with definitions for key terms used throughout the paper to enhance overall clarity. Notably, we have replaced 'level progress' with the more precise term 'level competence.' Additionally, we have opted for 'auxiliary reward' instead of 'intrinsic reward,' taking into consideration the valuable suggestions provided by the reviewers. We appreciate their insights into the choice of terminology.

We will establish definitions for each term herein.

**Level Competence**: An indicator of success in an episode of game play on a specific level, reaching 100% when the specific level is successfully solved in that episode. (Note: This is the term formerly referred to as 'level progress.')

**Global Competence**: A measure of success in an episode of game play with respect to the target level, reaching 100% if the target level is successfully solved in that episode.

**Level Advancement**: Difference between current level competence and initial level competence within a particular level in the curriculum; it is a measure of how much competence increases or decreases in the same level.

**Global Advancement**:  Difference between current global competence and initial global competence across levels in the curriculum; it is a measure of how much competence increases or decreases with respect to achieving the target level.

**Auxiliary reward**: A scaled level competence

We will respond to additional results on baselines in a separate comment due to word limit in the comment section.

---

### Author Response · Authors · 2023-11-19
**General comment to all reviewers (2/2) - additional results on baselines**

We thank all reviewers again for their time and valuable feedback on our paper. We have conducted three baselines for the RL experiments that we would like to bring to the attention of the reviewers.

First, we have conducted a baseline where environments are chosen randomly from 1-5 water lanes, without the curriculum. This is done for six trials/random seeds with a limit of 10M frames. We find that the agent makes very little learning progress over 10M frames. The return for the goal level (5 water lanes) is generally zero over the entire 10M frames. Moreover, the agent does not obtain much training return either. We hypothesize that this is a difficult exploration problem for the agent without a curriculum, and although the agent may eventually make progress, the timescale would far exceed the 10M frame budget we used for our experiments.

Second, we conducted a baseline against Section 5.3 (curriculum without auxiliary reward) where, 30% of the time, previously solved levels are chosen based on where the agent currently is in the curriculum. For example, if the current curriculum progress is 3.5 water lanes, then levels would be chosen from [1, 2] water lanes 30% of the time. This is done for 6 trials/random seeds with a limit of 10M frames. We do find that this mitigates catastrophic forgetting, as training divergence did not happen in any of the trials. However, the agent makes less progress towards the curriculum than in Section 5.4 with the auxiliary reward. In Section 5.4, all six trials got to at least 4 water lanes, whereas in this baseline, only one trial got to 4 water lanes. The average number of water lanes across all trials was 2.958. Additionally, in this baseline, none of the trials were able to solve the goal level (unlike in Section 5.4, where 5 of the trials could solve the goal level). We find that although stochastically selecting previously solved levels prevents catastrophic forgetting, it may not be an optimal approach for advancing the curriculum. We hypothesize that this occurs because 30% of the training budget is being spent to prevent agent forgetting, rather than learning how to solve the latest level in the curriculum. Therefore, it would be valuable to have an agent learn and utilize the auxiliary reward used in Section 5.4, as this both prevents catastrophic forgetting and advances the curriculum.

Third, we conducted a baseline that was similar to the second one, except instead of sampling previously solved levels, all possible levels (from 1 to 5 water lanes) were sampled. Six trials/random seeds were used with a limit of 10M frames. This baseline generally does poorly. In 4 trials, the training divergence still occurred. In only one trial was the training still active; this trial reached 3.0 water lanes. We hypothesize that this baseline performs worse than sampling previously solved levels, because sampling more difficult levels generally leads to more difficult levels that may not be solvable. This yields sparser returns and induces the same training divergence behavior seen in Section 5.3.

Additionally, we are actively planning to incorporate percentage groups derived from the child study into the RL agent study, aiming for a comprehensive integration of observations from children's behavior into the analysis of RL agents. We look forward to sharing our results and insights on these enhancements.

---

### Meta-Review · Area_Chair_SXaP · 2023-12-06

**Metareview:**

The paper conducts a user study with children to understand their learning strategies in tackling challenging tasks within Procgen environments. Then, observations from this study are used to investigate automated curriculum strategies for RL agents. The reviewers acknowledged that the paper provided interesting insights from the user study with children and appreciated the idea of designing curriculum strategies inspired by human learning. However, the reviewers pointed out several weaknesses in the paper and shared common concerns. We want to thank the authors for their detailed responses. Based on the raised concerns and follow-up discussions, unfortunately, the final decision is a rejection. Nevertheless, the reviewers have provided detailed and constructive feedback. We hope the authors can incorporate this feedback when preparing future revisions of the paper.

**Justification For Why Not Higher Score:**

The reviewers pointed out several weaknesses in the paper. There was a consensus among the reviewers that the work is not yet ready for publication.

**Justification For Why Not Lower Score:**

N/A

---

### Decision · Program_Chairs · 2024-01-16

Reject